# Alternative polyadenylation factor *CPSF6* regulates temperature compensation of the mammalian circadian clock

**Christoph Schmal** [1☉]*, **Bert Maier**[2☉], **Reut Ashwal-Fluss**[3], **Osnat Bartok**[3], **Anna-Marie Finger**[2], **Tanja Bange**[4], **Stella Koutsouli**[4], **Maria S. Robles**[4], **Sebastian Kadener**[3,5], **Hanspeter Herzel**[1], **Achim Kramer**[2]

**1** Institute for Theoretical Biology, Humboldt-Universität zu Berlin and Charité-Universitätsmedizin Berlin, Berlin, Germany, **2** Laboratory of Chronobiology, Institute for Medical immunology, Charité-Universitätsmedizin Berlin, Berlin, Germany, **3** Silberman Institute of Life Sciences, The Hebrew University of Jerusalem, Jerusalem, Israel, **4** Institute of Medical Psychology, Faculty of Medicine, Ludwig-Maximilians-Universität München, München, Germany, **5** Department of Biology, Brandeis University, Waltham, Massachusetts, United States of America

☉ These authors contributed equally to this work.
* christoph.schmal@hu-berlin.de

**Data Availability Statement:** All raw and processed sequencing data generated in this study have been submitted to the NCBI Gene Expression Omnibus (GEO; https://www.ncbi.nlm.nih.gov/geo/

## Abstract

A defining property of circadian clocks is temperature compensation, characterized by the resilience of their near 24-hour free-running periods against changes in environmental temperature within the physiological range. While temperature compensation is evolutionary conserved across different taxa of life and has been studied within many model organisms, its molecular underpinnings remain elusive. Posttranscriptional regulations such as temperature-sensitive alternative splicing or phosphorylation have been described as underlying reactions. Here, we show that knockdown of cleavage and polyadenylation specificity factor subunit 6 (*CPSF6*), a key regulator of 3′-end cleavage and polyadenylation, significantly alters circadian temperature compensation in human U-2 OS cells. We apply a combination of 3′-end-RNA-seq and mass spectrometry–based proteomics to globally quantify changes in 3′ UTR length as well as gene and protein expression between wild-type and *CPSF6* knockdown cells and their dependency on temperature. Since changes in temperature compensation behavior should be reflected in alterations of temperature responses within one or all of the 3 regulatory layers, we statistically assess differential responses upon changes in ambient temperature between wild-type and *CPSF6* knockdown cells. By this means, we reveal candidate genes underlying circadian temperature compensation, including eukaryotic translation initiation factor 2 subunit 1 (*EIF2S1*).

## 1. Introduction

Circadian clocks are heritable endogenous oscillators that have evolved convergently across different taxa of life as adaptions to environments with predictable daily changes, such as light intensity and temperature [1–3]. A defining feature of circadian clocks is that their rhythmicity

) under accession number GSE185896. The mass spectrometry proteomics data have been deposited to the ProteomeXchange Consortium via the PRIDE partner repository with the dataset identifier PXD029343. All other relevant data can be found within the Supporting information.

**Funding:** CS acknowledges support by the Deutsche Forschungsgemeinschaft (DFG, German Research Foundation) through grant number SCHM3362/2-1 (Project-ID 414704559). SK acknowledges support by the National Institute of Health under award number R01GM125859. Work in AK's and HH's laboratory is funded by the Deutsche Forschungsgemeinschaft (DFG, German Research Foundation) - Project-ID 278001972 - TRR 186. We thank the LMU Munich's Institutional Strategy LMUexcellent within the framework of the German Excellence Initiative and the Deutsche Forschungsgemeinschaft (DFG, German Research Foundation) SFB1321 (Project-ID 329628492) for funding the work of MSR. The funders had no role in study design, data collection and analysis, decision to publish, or preparation of the manuscript.

**Competing interests:** The authors have declared that no competing interests exist.

**Abbreviations:** CFI, cleavage factor I; CKI$\epsilon$, casein kinase I$\epsilon$; CKII, casein kinase 2; CPSF6, cleavage and polyadenylation specificity factor subunit 6; EIF2S1, eukaryotic translation initiation factor 2 subunit 1; ESAT, End Sequence Analysis Toolkit; NCOM, normalized center of mass; PAS, polyadenylation site; RNAi, RNA interference; RSA, Redundant SiRNA Activity; SCN, suprachiasmatic nucleus; shRNA, short hairpin RNA.

is self-sustained, even under constant environmental conditions, with free-running periods of approximately (but not exactly) 24 hours. Remarkably, this rhythmicity is generated cell-autonomously by molecular delayed interconnected negative feedback loops [4]. In mammals, the core loop relies on the rhythmic transcriptional activation of *Per* and *Cry* genes by CLOCK and BMAL1 heterodimers, followed by the antagonistic effect of PER and CRY proteins on their own transcription [5]. The interlocking of the core loop with additional positive and negative feedback loops is believed to give the system both plasticity and robustness [6–9]. While the role of posttranslational modifications of clock proteins is considered crucial for circadian dynamics [10], there is emerging evidence that also posttranscriptional pre-mRNA processing steps such as capping, splicing, polyadenylation, or nucleus–cytosol shuttling contributes to circadian gene expression [11].

Under natural conditions, environmental influences such as temperature, light intensity, or food availability can vary greatly with the time of day, but also at higher latitudes with season, and are subject to unpredictable fluctuations due to weather effects. Nevertheless, a functional clock system should reliably tell time even under such circumstances. Interestingly, although the rates at which chemical reactions occur generally increase substantially with temperature [12,13], circadian free-running periods have been found to be remarkably resilient to changes in ambient temperature [14]. This phenomenon, termed temperature compensation, is a defining feature of circadian clocks [15] that has been found in various kingdoms of life, ranging from poikilothermic organisms such as single-cell algae [16,17], higher plants [18], fungi [19,20], insects [21], and reptiles [22] to cells and tissues of homeothermic animals such as rodents [23,24] and human [25]. In 1973, Pittendrigh and Caldarola proposed that temperature compensation is only one aspect of an overarching homeostatic mechanism that protects the dynamics of the circadian clock from external influences [26] including metabolic fluctuations [27,28]. In addition, temperature compensation among other pacemakers such as lunar and semilunar clocks [29] has been reported, underpinning its importance for reliable time-telling even across different time scales.

Although temperature compensation of circadian rhythms was first described more than half a century ago, the underlying molecular mechanisms and regulatory elements involved remain poorly understood. Several temperature decompensation mutants exhibiting an altered temperature dependency of the circadian free-running period compared to wild type behavior have been described among various organisms. For example, the *tau* mutation that has been the first described single-gene mutation affecting the circadian free-running period in mammals [30] additionally leads to a temperature decompensation phenotype in retina cells with an increased value of $Q_{10} = 1.487$ compared to $Q_{10} = 1.096$ of wild-type golden hamster [31]. In *Drosophila melanogaster*, mutations in the dimerization domain (polyadenylation site (PAS)) of the core clock protein period (*per*) lead to a lengthening of the free-running period and an altered temperature compensation phenotype with a decreased $Q_{10} = 0.88$ compared to $Q_{10} = 0.98$ of the wild-type *Canton strain* (own calculation based on Table 1 of [32]). However, it should be noted that many but not every clock gene mutant that shows an altered circadian period additionally exhibits a temperature decompensation phenotype [33,34]. In addition, temperature dependencies (i.e., $Q_{10}$ values) of circadian free-running periods may nonlinearly depend and be specific to certain temperature ranges as shown for *Neurospora crassa*, where free-running periods of wild-type and several clock mutant strains are relatively resilient at lower temperatures and become more temperature-dependent at higher temperature values [19,20].

Interestingly, it has been reported that posttranscriptional and posttranslational mechanisms such as thermosensitive promoter use, alternative splicing, and control of translation rates or phosphorylation mediate both temperature compensation as well as temperature

responses [35–44]. For example, in *Neurospora crassa*, thermosensitive alternative splicing of the pre-mRNA of clock gene frequency (*frq*) influences the circadian period length [43,44]. In addition, alterations of casein kinase 2 (CKII) expression as well as mutations in putative CKII phosphosites of FRQ affect the temperature compensation of *Neurospora* [37]. Likewise, inhibition of casein kinase I$\epsilon$ (CKI$\epsilon$) or CKI$\delta$-dependent phosphorylation modulates the period of the mammalian circadian clock [45]. Interestingly, CKI$\epsilon$/$\delta$-dependent phosphorylation is insensitive to temperature changes [45] with a CKI-specific domain being even able to confer temperature compensation to noncompensated kinases [46]. Mutation of this domain significantly altered the temperature compensation of circadian clock period in the suprachiasmatic nucleus (SCN) of mice [46], suggesting that mammalian circadian temperature compensation is regulated by CKI$\epsilon$/$\delta$-dependent phosphorylation. This CKI-dependent phosphorylation activity appears to be essential even for temperature compensated circadian rhythms in the absence of the circadian transcription–translation feedback loop as inhibition of CKI abolishes temperature compensation of red blood cells as well [47]. Furthermore, a recent study identifies Na$^+$/Ca$^{2+}$ exchanger (NHX) mediated cold Ca$^{2+}$ signaling as an important regulator of mammalian temperature compensation that seems to be conserved in eukaryotes and prokaryotes [48].

Here, by using a systematic genetic screen, we found that knockdown of cleavage and polyadenylation specificity factor subunit 6 (*CPSF6*) leads to both long circadian periods and impaired circadian temperature compensation in human U-2 OS cells. *CPSF6* is a core component of the cleavage and polyadenylation machinery, a large multiprotein complex constituted by dozens of different proteins. Two subunits of NUDT21 (CPSF5 or CFI25) together with either CPSF6 (CFI68) or CPSF7 (CFI59) build the heterotrimeric complex cleavage factor I (CFI) that binds to UGUA regulatory *cis*-elements, enriched in upstream parts of distal polyadenylation sites [49]. Alternative polyadenylation that is highly gene and tissue specific [50] generates mRNA isoforms with different noncoding 3′ UTRs length and constitutes a major gene regulatory mechanism by altering the presence and accessibility of regulatory elements within 3′ UTRs for *trans*-acting molecules such as micro-RNA or RNA-binding proteins [51]. The importance of CPSF6 for the utilization of distal polyadenylation sites and thus for normal lengths of 3′ UTRs together with the fact that CPSF5 phenocopies the behavior of CPSF6 in terms of temperature compensation suggests that a thermosensitive use of alternative polyadenylation sites might be a possible mechanism underlying circadian temperature compensation in the human circadian clock. Using a global integrative multiomics analysis comparing the differential responses upon adiabatic temperature changes between wild-type and *CPSF6* knockdown cells at the level of 3′ UTR length, transcript, and protein expression, we identified candidate genes at the core of circadian temperature compensation.

## 2. Results

### 2.1. CPSF6 regulates circadian temperature compensation

Temperature compensation is defined by the relative resilience of the circadian free-running period with respect to temperature changes. Thus, depletion of essential components underlying temperature compensation should lead to a circadian period phenotype in almost all cases, i.e., at all temperatures apart from those particular cases where the curves describing the free-running period versus temperature relationship of wild-type and temperature decompensated cells cross each other. Thus, to search for genes that regulate circadian temperature compensation, we performed a large-scale RNA interference (RNAi) screen for aberrant circadian period phenotypes in human U-2 OS cells, an established model of the mammalian circadian clock [52,53]. Posttranscriptional mechanisms have been supposed to mediate temperature

responses and temperature compensation among different organisms such as *Neurospora* [35,37], *Drosophila* [36,41], or *Arabidopsis* [38,39]. Therefore, we depleted the transcripts of 1,024 genes associated with posttranscriptional regulation and RNA processing [54] in U-2 OS reporter cells with RNAi constructs from our laboratory library and analyzed the circadian *Bmal1:Luc* bioluminescence rhythms at 35˚C as reported previously [52,53]. Knockdown of 83 genes showed significant effects on circadian periods ($p<0.01$; S1 Table), and depletion of *CPSF6* resulted in the strongest phenotype. We found this phenotype to be dependent on the strength of *CPSF6* mRNA expression where an increasing knockdown strength leads to longer periods (S1A Fig), resulting in a dose-dependent overall period lengthening of approximately 1.5 to 2 hours in both human U-2 OS (Figs 1A, S1A and S1B) and mouse NIH3T3 cells (S1C and S1D Fig). *CPSF6* is part of a multiprotein complex that is important for 3′-end processing of transcripts defining the polyadenylation site and, thus, the 3′ UTR length. Interestingly, knockdown of *NUDT21* (also known as *CPSF5*) that builds together with either *CPSF6* or *CPSF7* the heterotrimeric CFI polyadenylation complex phenocopies the effect of *CPSF6* knockdown (S1E and S1F Fig), while no significant period phenotype ($\tau = 23.77$h, $p = 0.48$) can be found for a knockdown of *CPSF7* in our large-scale RNAi screen. Similarly, knockdown of *CPSF6* and *NUDT21* but strikingly not *CPSF7* leads to a global shortening of 3′ UTRs [55,56], suggesting that *CPSF6* acts on the circadian clock via its role within the polyadenylation complex.

Since 3′ UTRs contribute to the rhythmic expression of clock and clock-controlled genes [57,58] and thermosensitive regulation of pre-mRNA processing has been discussed as a mechanism underlying temperature sensation and compensation in other organisms [36,43,44], we investigated the effect of temperature on free-running periods in *CPSF6* knockdown cells. Interestingly, knockdown of *CPSF6* significantly affects temperature compensation. Whereas the free-running periods of control cells changed only slightly from 24.4h±0.3h at 32˚C to 23.5h±0.4h at 39˚C, the temperature dependence was significantly ($p<0.001$, statistical test for differential slopes, see S2A Fig and Materials and methods) altered upon *CPSF6* knockdown with a shortening of free-running period from 27.4h±0.3h at 32˚C to 24.2h±0.6h at 39˚C (Fig 1B). These dependencies led to an increase of the period temperature coefficient from $Q_{10} = 1.06\pm0.01$ to $Q_{10} = 1.19\pm0.02$ upon *CPSF6* knockdown (S2A Fig). Importantly, this phenotype is not a consequence of a temperature-dependent short hairpin RNA (shRNA)-mediated knockdown efficiency as the relative *CPSF6* expression in control and *CPSF6*-depleted cells is resilient to temperature changes (Fig 1C). In addition, knockdown of *NUDT21* phenocopies the effect of *CPSF6* depletion on temperature compensation of the circadian period in U-2 OS (S2B Fig), again suggesting that *CPSF6* acts on the circadian clock via its role within the polyadenylation complex.

**2.1.1. CPSF6 knockdown alters mRNA stability and expression of clock genes.** It has been described that *CPSF6* knockdown induces a systematic shift toward proximal PASs and thus to shorter 3′ UTRs [59]. To investigate, whether *CPSF6* knockdown has an impact on 3′ UTR length of canonical clock genes, we measured the expression of long 3′ UTR isoforms versus the total amount of mRNA (Figs 1D–1F and S3). Among the 9 clock genes examined, namely *ARNTL*, *CRY1/2*, *CLOCK*, *NR1D1/2*, and *PER1/2/3*, transcripts of *PER3* showed the largest and most significant ($p_i = 0.003$) shift from distal to proximal PAS usage at 37˚C, followed by *CLOCK* and *NR1D2* (Fig 1F *right*). Whereas *CRY2*, *NR1D1*, as well as *PER1* showed no significant change in polyadenylation-site usage, *CPSF6* knockdown resulted in a small increase of distal PAS usage for *CRY1* and *PER2* (S3B Fig, right panel).

Since 3′ UTRs have been shown to affect mRNA stability [51], we compared the mRNA half-lives of clock genes in unsynchronized control and *CPSF6* knockdown cells at 37˚C (Fig 1E and 1F, left), using the transcription inhibitor *triptolide* [60]. For *CLOCK* and *NR1D2*,

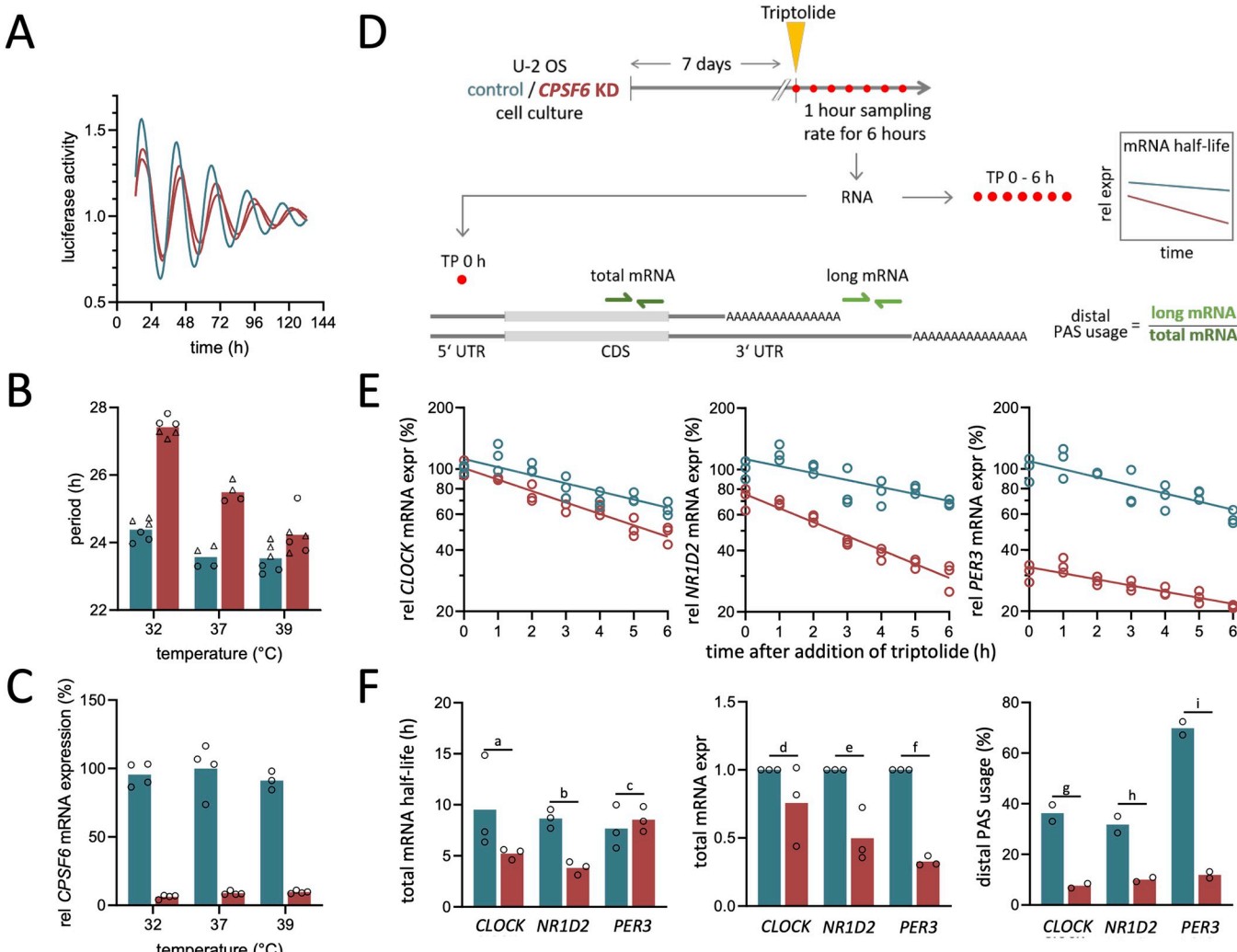

**Fig 1. Knockdown of *CPSF6* impairs circadian temperature compensation.** (**A**) Circadian free-running period of U-2 OS cells lengthens upon shRNA-mediated *CPSF6* knockdown (red) in comparison with control cells (blue), as accessed by a *Bmal1-luciferase* reporter construct. Raw time series data have been normalized by its baseline trend (magnitude). (**B**) Temperature compensation is significantly ($p<0.001$; $n = 32$; see S2A Fig for details) reduced upon *CPSF6* knockdown. (**C**) Relative *CPSF6* mRNA expression ($n = 3–4$ per condition) in control (blue) as well as *CPSF6*-depleted (red) cells at 3 different temperatures. Expression values are normalized to the average expression of control cells at 37°C. (**D**) Schematic drawing of experimental setup for determination of alternative PAS usage, transcript expression, as well as transcript half-lives. (**E**) Dynamics of *CLOCK* (left), *NR1D2* (middle), and *PER3* (right) mRNA expression after application of transcription inhibitor *triptolide* in wild-type (blue) and *CPSF6* knockdown cells (red). Straight lines depict linear fits to the logarithmic data as used for the determination of half-lives. (**F**) Bar plots, comparing the total mRNA half-life (left) as determined from the fits in panel (**E**), total mRNA expression (middle), and distal PAS usage (right) in *CLOCK*, *NR1D2*, and *PER3* in wild-type versus *CPSF6* knockdown cells. Each dot denotes results from an individual experiment where values of 3 technical replicates have been averaged. Bars denote the corresponding averages of these individual experiments. Please note that results in panel (E) show dynamics of the 3 technical replicates within 1 individual experiment, thus corresponding to 1 dot in panel F (left). $P$ values for a statistical comparison (*t* test) between the total mRNA half-life, total mRNA expression, and distal PAS usage between control and *CPSF6*-depleted cells are $p_a =0.187$, $p_b = 0.002$, $p_c = 0.580$, $p_d = 0.225$, $p_e = 0.012$, $p_f<0.001$, $p_g = 0.014$, $p_h = 0.023$, $p_i =0.003$. Raw data underlying panels A, B, C, E, and F can be found in S1 and S5 Data. *CPSF6*, cleavage and polyadenylation specificity factor subunit 6; PAS, polyadenylation site; shRNA, short hairpin RNA.

which showed a significant shift toward shorter 3′ UTRs, we observed reduced mRNA half-lives (Fig 1E and 1F, left), in contrast to the expectation that a lack of miRNA binding sites in transcripts with short 3′ UTRs generally leads to an increased stability [61]. However, *PER3*, which also had an overall shorter 3′ UTR in *CPSF6* knockdown cells (Fig 1F, right), showed no changes in mRNA stability (Fig 1E and 1F, left). In all 3 cases, *CPSF6* knockdown reduced

total mRNA expression ([Fig 1F](), center). Thus, whereas shortening of the 3′ UTR length in *CLOCK* and *NR1D2* results in shorter transcript half-lives, presumably through a loss of regulatory elements within the UTR, our data suggest that processes other than mRNA stability regulate changes of *PER3* expression.

Of note, we did not detect a systematic temperature effect on the 3′ UTR lengths of clock genes, nor was the *CPSF6* knockdown effect on the 3′ UTR lengths of the clock genes examined temperature dependent. In order to further investigate if CPSF6 affects temperature compensation via alternative polyadenylation within the core clock machinery, we studied the impact of the 3′ UTR length of core clock gene *CLOCK* on circadian parameters. Here, we focused on *CLOCK* as it has the longest 3′ UTR among all canonical clock genes, it is essential for a normal circadian phenotype [62], and it shows a large shift towards shorter 3′ UTRs upon *CPSF6* depletion that is accompanied by a decrease in its mRNA half-life and total mRNA expression ([Fig 1F]()). By using a CRISPR-Cas9–mediated depletion of the *CLOCK* 3′ UTRs PASs, we could show that the length of the *CLOCK* 3′ UTR neither regulates the circadian period nor does it phenocopy the *CPSF6* knockdown period lengthening phenotype (see [S1 Text]()). Taken together, this suggests that the role of *CPSF6* in circadian temperature compensation is not limited to the core oscillator, but may be a higher-level mechanism. To search for regulatory elements outside the canonical set of core clock genes, we employ a combination of different high-throughput analysis methods.

## 2.2. *CPSF6* knockdown induces global changes in polyadenylation site usage, transcript, and protein expression

The increased temperature sensitivity of free-running periods among *CPSF6* knockdown cells suggests regulatory interactions at the pre-mRNA processing level as a potential mechanism underlying circadian temperature compensation in mammals. By combining transcriptomics, proteomics, as well as RNAi high-throughput approaches, we aimed to uncover key regulatory components that contribute to the observed differential temperature response between *CPSF6* knockdown and wild-type cells and thus temperature compensation. To investigate the impact of temperature and *CPSF6* on PAS selection, total transcript, and protein expression, we employed an integrative omics approach in *CPSF6* knockdown and control cells at 3 different temperatures (32°C, 37°C, and 39°C) ([Fig 2A]()). Below, we first describe the effect of *CPSF6* knockdown on PAS usage, transcript, and protein abundance at one temperature (37°C), before reporting whether and to what extent temperature modulates these effects. Please note that for visual purposes in figure organization, in the following, subpanel ordering of Figs [2]() and [3]() does not always correspond with the first reference in the main text.

**2.2.1. Polyadenylation site usage is globally controlled by CPSF6.** As a prerequisite for further analyses, we first precisely mapped PASs of transcripts in U-2 OS cells by means of an RNaseH-seq approach as previously described [63]. Of all ≈20.000 genes for which we found polyadenylation signals, the vast majority of 73.4% contained multiple polyadenylation signals with 52.6% having 3 or more ([S4 Fig]() and [S2 Table]()), similar to what has been reported for other tissues and cell types [64,65]. To investigate the impact of *CPSF6* on PAS usage, we performed 3′-end RNA-seq and calculated the "center of mass" of the read distribution along the annotated 3′ UTR, normalized to the lengths of the 3′ UTR. Hereafter, we refer to this quantity as the normalized center of mass (NCOM; see [Fig 2A]()) that takes values of 0 and 1 in the extreme cases of all reads aligned to the starting and end points of the 3′ UTR, respectively, and intermediate values for all other cases. Knockdown of *CPSF6* resulted in a global shift toward use of proximal polyadenylation sites leading to overall shorter 3′ UTRs (Figs [2B]() and [S5A]() and [S3 Table]()). A significant shift of the polyadenylation site was observed in approximately one-third

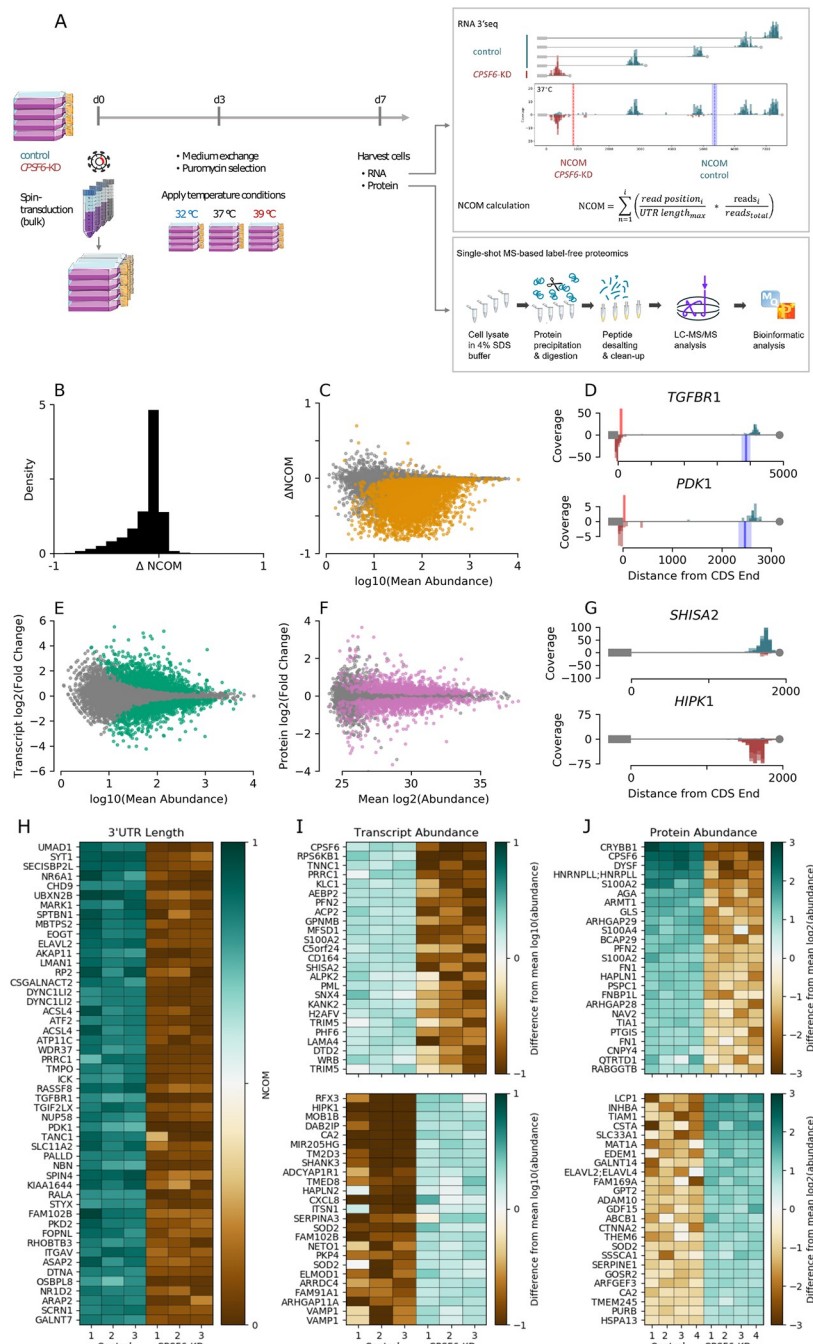

**Fig 2. Knockdown of *CPSF6* leads to global shifts in PAS usage as well as major transcript and protein abundance changes.** (**A**) Schematic of the integrative omics experimental approach used throughout this study. (**B**) Histogram depicting the absolute shift in NCOM (i.e., normalized 3′ UTR length) upon *CPSF6* knockdown at 37˚C. (**C**) Absolute shift in NCOM upon *CPSF6* knockdown versus decadic logarithm of the corresponding mean expression across both genetic backgrounds. Isoforms showing a significant NCOM shift (*t* test, *n* = 3 technical replicates per condition) are depicted by orange dots (Benjamini–Hochberg corrected *p*<0.05). (**D**) Two exemplary 3′ UTR read distributions of genes showing a significant shortening of 3′ UTR length upon *CPSF6* knockdown (red) in comparison to wild type (blue). Note that the 3 technical replicates are overdubbed. Bold vertical lines and attached shaded areas denote the NCOM average and standard deviation among the replicates, respectively. (**E**) MA plot showing the binary logarithm of the fold change between wild-type and *CPSF6* knockdown transcript expression versus the decadic logarithm of the corresponding mean expression. Statistically significant differential expressions as determined via DESeq are denoted by green markers (Benjamini–Hochberg corrected *FDR*<0.05). (**F**) MA plot showing the binary logarithm of fold changes between wild-type and *CPSF6* knockdown cells versus corresponding mean protein LFQ intensities, which we

will thereinafter term protein abundances for the sake of simplicity. Proteins showing a significantly different abundance (*t* test, *n* = 4 technical replicates per experimental condition) are depicted by pink dots (Benjamini–Hochberg corrected *p*<0.05). (**G**) Representative examples that show a decrease (*SHISA2*, top) or increase (*HIPK1*, bottom) of transcript abundance upon *CPSF6* knockdown. (**H**) Heatmap depicting NCOM values for the 50 isoforms with the largest shift among those transcripts that show a significant NCOM change upon *CPSF6* knockdown. (**I**) Heatmap showing 25 transcript isoforms with a significant negative (top) and positive (bottom) fold change upon *CPSF6* knockdown, sorted by the amount of fold change. (**J**) Same as panel (**I**) for protein LFQ abundance. Data underlying panels A-E, G-I, and F, J is available from the NCBI (http://www.ncbi.nlm.nih.gov/geo/) under accession number GSE185896 and the ProteomeXchange Consortium via the PRIDE partner repository (http://www.ebi.ac.uk/pride/archive/) with the dataset identifier PXD029343, respectively, and is contained in S3–S5 Tables. CPSF6, cleavage and polyadenylation specificity factor subunit 6; GEO; Gene Expression Omnibus; NCOM, normalized center of mass; PAS, polyadenylation site.

of all ≈12,000 isoforms detected (Figs 2C and S5B, orange dots), with the majority (99%) expressing a short 3′ UTR transcript variant, which is very similar to previous reports [55,56,59]. This results in a reduced overall median 3′ UTR length of approximately 428 nucleotides in *CPSF6*-depleted cells as compared to a median of 715 nucleotides in control cells (S5C and S5D Fig). Fig 2D illustrates 2 representative examples of read distributions of transcripts showing a significant switch from expression of long 3′ UTR isoforms in wild type (blue) to expression of short 3′ UTR isoforms in *CPSF6* knockdown cells (red), while Fig 2H shows a heatmap depicting values of the 50 isoforms with the largest NCOM shift in the set of genes showing a significant alteration in 3′ UTR length.

**2.2.2. Transcript and protein abundance are globally controlled by CPSF6.** To study the functional consequences of alternative polyadenylation, we next quantified changes in gene expression after *CPSF6* knockdown. We detected 1,924 (16%) differentially expressed transcripts (Fig 2E, green dots; Fig 2G shows 2 examples), of which 977 were up-regulated and 947 were down-regulated (Fig 2I and S4 Table). Interestingly, these transcripts were enriched within the group of transcripts that also exhibited a significant shift in 3′ UTR length (Fig 3A, light green), indicating a dominant role of 3′ UTR length in gene expression, as previously reported [51]. Out of the 967 transcripts that differed significantly at both levels, 60% showed an increase and 40% a decrease in gene expression, respectively. This is consistent with recent findings reporting that a reduction in 3′ UTR length can lead to both increased or decreased gene expression [51], in contrast to the previously prevailing assumption that a shorter 3′ UTR generally leads to an increased mRNA stability by loosing miRNA binding sites.

To gain further insight into the functional implications of *CPSF6* knockdown, we globally examined how *CPSF6*-dependent shortening of 3′ UTRs affects protein abundance (Fig 2A). Using mass spectrometry–based quantitative proteomics (S6 Fig), we quantified 4,733 proteins across all samples. Of these, 1,683 (36%) showed a significantly different abundance after *CPSF6* knockdown (Fig 2F, pink), with 50% up-regulated and 50% down-regulated (Fig 2J and S5 Table). These altered protein-coding genes were significantly enriched in the group of genes that also showed significant regulation at the transcriptional level (Fig 3B), indicating a high correlation between mRNA and protein expression, as previously described [66]. Although still significant, genes whose protein level was significantly altered upon *CPSF6* knockdown were enriched to a much lesser extent in the set of transcripts that exhibited a change in 3′ UTR length after *CPSF6* knockdown (Fig 3C), indicating a relatively weaker effect of global 3′ UTR shortening on protein abundance, similar to what has been previously described for human and murine T cells [67].

In summary, knockdown of *CPSF6* induces global changes in 3′ UTR length as well as in transcript and protein abundance at physiological temperatures of 37˚C (Fig 3D).

## 2.3. Temperature has little effect on global alternative polyadenylation but major influence on transcript and protein abundance

To identify genes that control circadian temperature compensation in wild-type cells or analogously promote temperature decompensation in *CPSF6* knockdown cells, we next examined

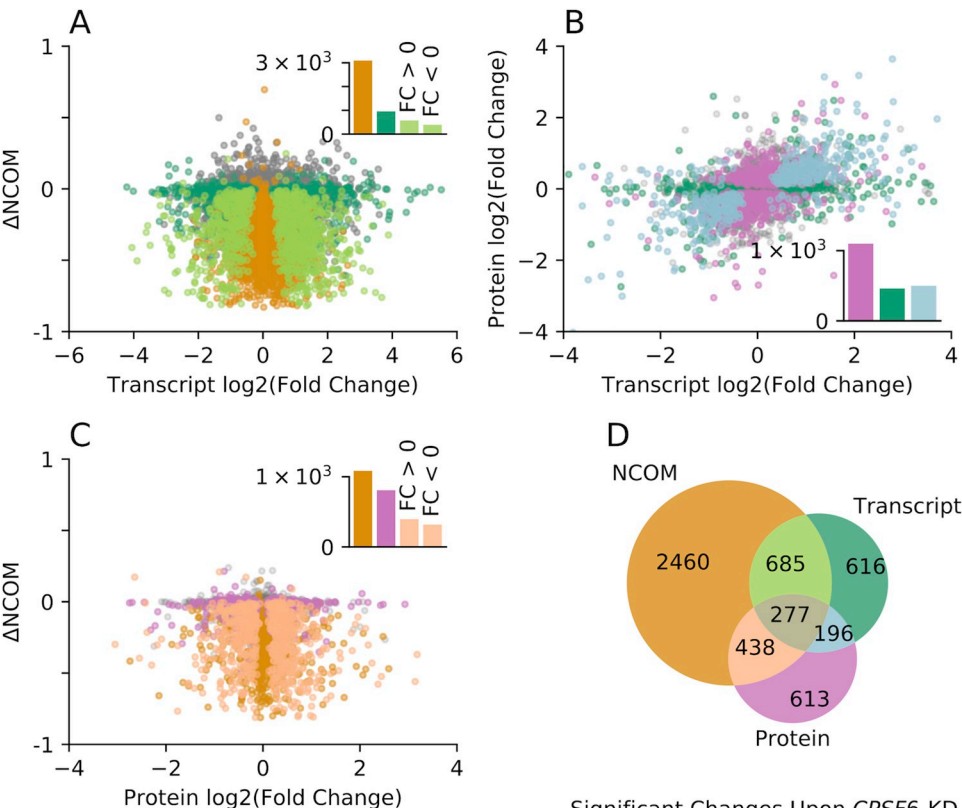

**Fig 3. Global shortening of 3′ UTR lengths upon *CPSF6* knockdown induces global changes in transcript and protein abundance.** (**A**) Scatter plot depicting the shift in 3′ UTR lengths (ΔNCOM) versus binary logarithm of fold changes in transcript expression upon *CPSF6* knockdown at 37°C. Δ*NCOM* is defined as the normalized center of mass (NCOM) in wild-type U-2 OS cells, subtracted from the NCOM in *CPSF6* knockdown cells. Different colors denote transcripts with no significant changes in NCOM and transcript abundance (gray), significant changes exclusively at the NCOM (orange) or transcript (dark green) level, or at both levels at the same time (light green), respectively. Transcripts with a significant fold change upon *CPSF6* knockdown are enriched within the set of transcripts that show a significant shift in 3′ UTR length (Fisher's exact test: $p<0.001$, odds ratio 2.24). (**B**) Binary logarithm of protein fold changes upon *CPSF6* knockdown are depicted versus corresponding binary logarithms of fold changes of transcript expression. Different colors denote genes with no significant changes in transcript and protein expression (*gray*), solely in transcript abundance (dark green), solely in protein abundance (pink), or in transcript and protein abundance (light blue), respectively. Genes with a significant fold change at the protein level are significantly enriched within the set of genes that exhibit a significant fold change of transcript expression (Fisher's exact test: $p<0.001$, odds ratio 2.38). (**C**) Shift in 3′ UTR lengths (ΔANCOM) versus binary logarithm of fold change in protein abundance upon *CPSF6* knockdown. Different colors denote genes with no significant change in 3′ UTR length and protein abundance (gray), solely in 3′ UTR length (orange), solely in protein levels (pink), or a significant change in both (light orange), respectively. Proteins whose intensities are significantly different between conditions (fold change) are moderately enriched within the set of transcripts exhibiting a significant change in 3′ UTR length upon *CPSF6* knockdown (Fisher's exact test: $p = 0.0002$, odds ratio 1.27). (**D**) Venn diagram indicating the number of genes that show a significant alteration of 3′ UTR length (orange), transcript expression (green), or protein abundance (pink) upon *CPSF6* knockdown. Note that significance thresholds identical to those in Fig 2 have been used. Insets in panels A–C depict the absolute number of isoforms in each corresponding group. Data underlying this figure are available from the NCBI (http://www.ncbi.nlm.nih.gov/geo/) under accession number GSE185896 and the ProteomeXchange Consortium via the PRIDE partner repository (http://www.ebi.ac.uk/pride/archive/) with the dataset identifier PXD029343 and are contained in S3–S5 Tables. CPSF6, cleavage and polyadenylation specificity factor subunit 6; GEO; Gene Expression Omnibus; NCOM, normalized center of mass.

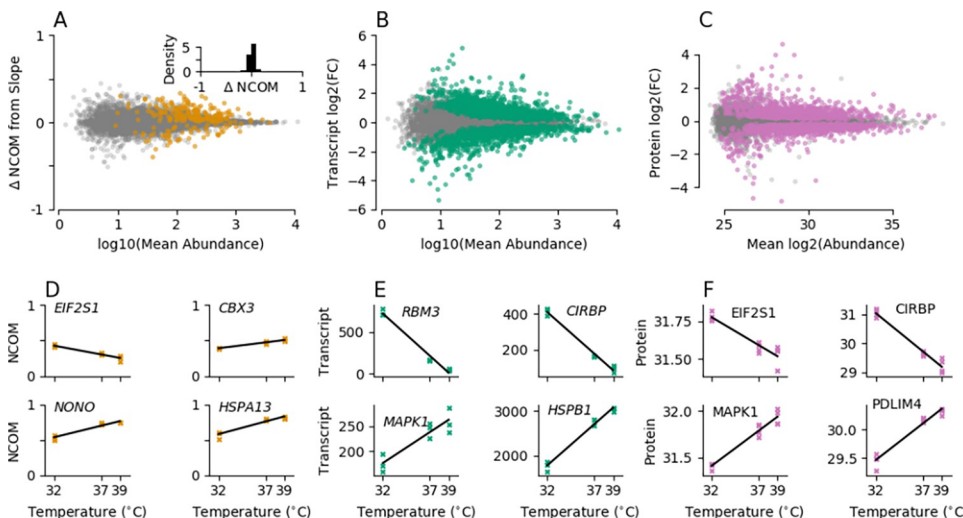

**Fig 4. Alterations in environmental temperature induce major expression changes while having a weak effect on 3′ UTR length.** (**A**) MA plot showing the shift in 3′ UTR length upon temperature variations between 32°C and 39°C as determined from a linear regression versus the decadic logarithm of the mean expression among all temperatures from the corresponding transcript. Transcripts with a statistically significant change in 3′ UTR length at a 5% FDR (Benjamini–Hochberg corrected *P* values) are depicted by orange markers. No global shift in 3′ UTR length can be observed upon temperature alterations in wild-type cells (inset). (**B**) Same as panel (**A**), showing the binary logarithm of transcript expression fold changes upon temperature variations between 32°C and 39°C on the ordinate. Transcripts with a statistically significant change in expression at a 5% FDR (Benjamini–Hochberg corrected *P* values) are depicted by green markers. (**C**) Same as panel (**B**), showing the binary logarithm of protein LFQ intensity fold changes upon temperature variations between 32°C and 39°C on the ordinate as well as the binary logarithm of the mean among all temperatures from the corresponding proteins. Proteins with a statistically significant change in abundance at a 5% FDR (Benjamini–Hochberg corrected *P* values) are depicted by pink markers. (**D-F**) 3′ UTR length (D, orange markers), transcript expression (E, green markers), and protein intensities (F, pink markers) in wild-type cells at all 3 experimentally measured temperatures for different example genes that show a statistically significant corresponding temperature response. Fits from a linear regression are depicted by black lines. Data underlying this figure are available from the NCBI (http://www.ncbi.nlm.nih.gov/geo/) under accession number GSE185896 and the ProteomeXchange Consortium via the PRIDE partner repository (http://www.ebi.ac.uk/pride/archive/) with the dataset identifier PXD029343 and are contained in S6–S8 Tables. FDR, false discovery rate; GEO; Gene Expression Omnibus; NCOM, normalized center of mass.

the extent to which temperature induces genome-wide changes in PAS usage as well as transcript and protein abundance. Interestingly, only minor alterations in PAS selection were observed upon changes in temperature in wild-type cells, especially compared to the substantial global shifts seen upon *CPSF6* knockdown (compare Figs 2B, 2C and 4A). Of the ≈12,000 isoforms analyzed, only 2% showed a significant temperature-mediated change in 3′ UTR length (Figs 4A, orange, and S7A and S6 Table). Of these, >75% showed an increase in NCOM with increasing temperatures, indicating that higher temperatures tend to elongate 3′ UTRs in U-2 OS cells. This is in contrast to previous reports that cold shock induces a switch towards longer 3′ UTRs in mouse embryonic fibroblasts [68].

Despite the relatively weak effect on 3′ UTR lengths, changes in ambient temperature resulted in major expression changes at the transcript and protein levels (Figs 4B, 4C, S7B and S7C and S7 and S8 Tables). A total of 2,445 (20%) transcripts showed a significantly different abundance upon temperature changes (Fig 4B and S7 Table). Of these, more than half (60%) showed an increased abundance at higher temperatures. Furthermore, 43% of the proteins show significant abundance changes across different temperatures with almost equal increases and decreases in abundance at higher temperatures (Fig 4C and S8 Table). Representative examples of isoforms showing a significant shift in 3′ UTR length or in transcript and protein abundance upon increasing temperatures from 32°C to 39°C are depicted in Fig 4D–4F,

respectively. A similar picture can be drawn for *CPSF6* knockdown cells in response to different ambient temperatures. As in wild-type cells, only a small fraction (1%) of transcripts had significantly different 3′ UTR lengths, whereas abundance changed substantially at the transcript (22%) and protein (50%) level (S8 Fig).

Taken together, these data suggest that the cellular response to temperature is largely independent of *CPSF6* at the global level. However, even though we do not observe a global shift in PAS selection upon temperature changes, we cannot exclude the possibility that temperature and *CPSF6* modulate 3′ UTR length and thus the expression of a master regulator of transcription and/or translation.

## 2.4. Toward identifying key players underlying temperature compensation

The temperature decompensation phenotype in *CPSF6*-depleted cells can be explained by a differential response to changes in ambient temperatures compared to the wild type behavior. To search for genes in our multiomics data set that show a significantly different response upon changes in temperature in 3′ UTR length as well as in transcript or protein abundance between wild-type and *CPSF6* knockdown cells, we apply an integrative statistical analysis. Analogous to the analysis in Figs 1B and S2, we assumed a linear relationship between the observed variable of interest (i.e., 3′ UTR length, transcript, or protein abundance) and temperature and tested whether the slope describing this linear temperature dependency is significantly different in wild-type and *CPSF6* knockdown cells (see section Materials and methods). We then converted the resulting slopes to $Q_{10}$ temperature coefficients for comparison with previous results.

Comparing wild-type with *CPSF6* knockdown cells, we observed a differential response to temperature for 182 transcripts (2%; Fig 5A, orange dots, and S9 Table) in terms of 3′ UTR length and for 942 transcripts (10%; Fig 5B, green dots, and S10 Table) in terms of expression level. At the protein level, we detected 678 proteins (14%; Fig 5C, pink dots, and S11 Table) whose temperature-dependent abundance differed between wild-type and knockdown cells. Given the strong global effect of *CPSF6* on PAS selection, the easiest hypothesis explaining the observed temperature decompensation in *CPSF6* knockdown cells would be that the differential response in PAS selection of these cells compared to wild type behavior translates into a differential response at the transcript level and then in turn at the protein level and by this causes the observed phenotype. Interestingly, only 6 genes (*CBX3*, also known as *HP1γ*; *EIF2S1*; *HSPA13*; *MAPK1*, also known as *ERK2*; *NAP1L1*; and *PDLIM4*) showed a significantly differential temperature response between wild-type and *CPSF6* knockdown cells at all 3 regulatory levels examined, i.e., 3′ UTR length, transcript abundance, and protein abundance (Figs 5D, 5E and S9).

If any of these genes contribute to the loss of temperature compensation of the circadian free-running period in *CPSF6* knockdown cells they should fulfill the following criteria: (1) genetic depletion should result in a circadian phenotype similar to that of *CPSF6* knockdown cells (2) such phenotype should depend on temperature (3) under the assumption that target 3′ UTR binding sites of these candidate genes lie downstream of PASs used in control but not *CPSF6* knockdown cells, a genetic depletion of these candidate genes in *CPSF6*-depleted cells is not expected to lead to additional effects on the circadian phenotype (4) if such phenotype relies on temperature-dependent alternative polyadenylation, impairment of the polyadenylation signals within the corresponding 3′ UTRs should lead to a circadian phenotype similar to that of *CPSF6* knockdown cells. Although a complete investigation of above criteria is beyond the scope of this paper, we used results of the initial RNAi screen to examine which genes identified by our differential response analysis also showed a period lengthening or shortening phenotype after RNAi-mediated knockdown. Of the 6 genes that exhibited a differential

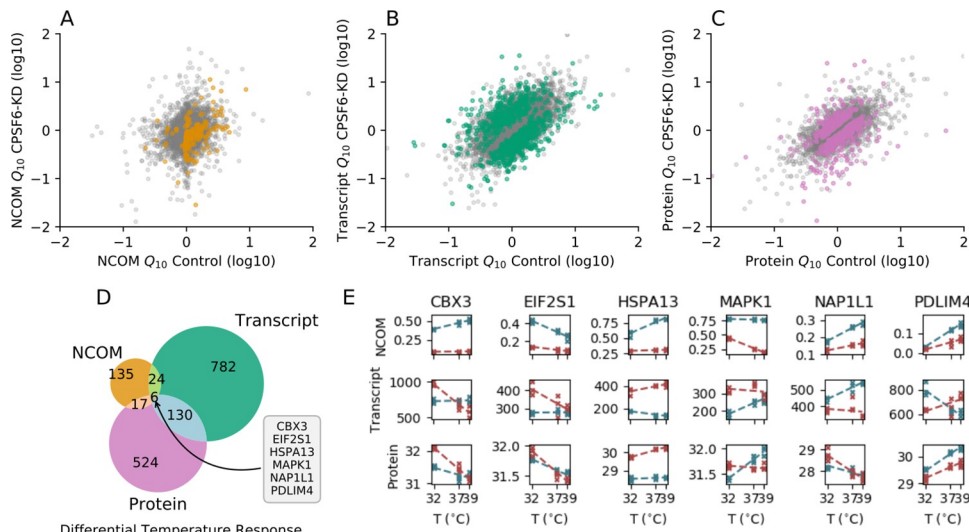

**Fig 5. Differential temperature responses between wild-type and *CPSF6* knockdown cells suggest candidates underlying temperature compensation.** (**A**) NCOM temperature coefficient $Q_{10}$ in *CPSF6* knockdown versus wild-type cells. Significantly differential responses in PAS selection (i.e., NCOM shifts) upon temperature changes between wild-type and *CPSF6* knockdown cells are depicted by orange dots. (**B**) Transcript expression temperature coefficient $Q_{10}$ in *CPSF6* knockdown versus wild-type cells. Significantly different responses in transcript expression upon temperature changes between wild-type and *CPSF6* knockdown cells are depicted by green dots. (**C**) Protein abundance temperature coefficient $Q_{10}$ in *CPSF6* knockdown versus wild-type cells. Significantly different responses in protein expression upon temperature changes between wild-type and *CPSF6* knockdown cells are depicted by pink dots. (**D**) Venn diagram indicating the number of genes that show a statistically significant differential temperature response between wild-type and *CPSF6* knockdown cells. (**E**) 3′ UTR length (top panels), transcript expression (middle panels), as well as protein abundance (lower panels) for 3 different temperatures in wild-type (blue) and *CPSF6* knockdown cells (red). Linear fits are denoted by dashed lines. Depicted are all 6 genes from panel (**D**) that show a differential temperature response between wild-type and *CPSF6* knockdown cells at all 3 regulatory layers. Data underlying this figure are available from the NCBI (http://www.ncbi.nlm.nih.gov/geo/) under accession number GSE185896 and the ProteomeXchange Consortium via the PRIDE partner repository (http://www.ebi.ac.uk/pride/archive/) with the dataset identifier PXD029343 and are contained in S9–S11 Tables. CPSF6, cleavage and polyadenylation specificity factor subunit 6; GEO; Gene Expression Omnibus; NCOM, normalized center of mass; PAS, polyadenylation site.

temperature response at all 3 regulatory levels, only *EIF2S1* (S10 Fig, top left) and, to a much weaker extent, *HSPA13* (S10 Fig, top middle) showed a significant period change after RNAi-mediated knockdown. To test whether knockdown of *EIF2S1* also leads to impaired temperature compensation, similar to what has been observed with *CPSF6* knockdown, we compared the circadian periods at 3 different temperatures via an additional small-scale RNAi experiment (S11Fig). Interestingly, knockdown of *EIF2S1* leads to a significant period increase in 2 out of 3 temperatures ($p<0.01$, *t* test) with periods similar to those of *CPSF6*-depleted cells (S11A Fig), thus fulfilling criterium one as discussed above. In addition, even though nonsignificant based on our differential response analysis ($p≤0.2$), knockdown of *EIF2S1* also resulted in temperature decompensation and an increase of the period temperature coefficient from $Q_{10} = 1.11±0.03$ to $Q_{10} = 1.20±0.07$ (S11B Fig), being consistent with criterium 2. Taken together, our results suggest that the role of *CPSF6* in circadian temperature compensation may be mediated via regulation of *EIF2S1*. Along these lines, a temperature-dependent differential expression of isoforms with different 3′ UTR lengths may be responsible for buffering the corresponding transcript and protein abundance against temperature changes. In line with this assumption, predominant temperature insensitive expression of the short 3′ UTR isoform in *CPSF6* knockdown cells leads to a stronger temperature dependency of *EIF2S1* transcript and protein abundance (5E Fig, second column).

## 3. Discussion

In all organisms with circadian clocks, the period of circadian oscillations is resilient to changes in ambient temperature, although the speed of chemical reactions is temperature dependent [14]. This temperature compensation is a defining feature of circadian clocks [15] and has been proposed to be part of a general homeostatic mechanism that confers robustness to the circadian timing system under a wide range of unstable environmental conditions [26–28]. However, the molecular mechanisms underlying circadian temperature compensation are still poorly understood.

Using a large-scale RNAi screen, we identified *CPSF6* (also known as *CFIm68*) as a regulator of circadian temperature compensation. *CPSF6* is a core component of the cleavage and polyadenylation machinery required for proper posttranscriptional processing of mRNA molecules. Interestingly, posttranscriptional regulation in mediating temperature perception by circadian clocks has been reported in several organisms. In *Drosophila*, thermosensitive alternative splicing of the *period* gene plays a critical role in the adaption to seasonally cold days [36]. Similarly, temperature-associated alternative splicing coupled to nonsense-mediated decay regulates the expression of several core clock genes in *Arabidopsis thaliana* [39,69]. In addition, temperature affects translational efficiency of certain plant genes such as bHLH transcription factor PIF7 through thermosensitive conformational changes within its mRNA 5′ UTR, thus influencing thermomorphogenesis [42]. Furthermore, posttranslational protein modifications such as phosphorylation or sumoylation have been shown to be important regulators of temperature compensation in plants, mammals, and fungi [37,38,70,71].

For our RNAi screen, we therefore focused on genes associated with posttranscriptional modification and RNA processing and identified *CPSF6* as the gene with the most significant period phenotype among all those tested. At a physiological temperature of 37°C, knockdown of *CPSF6* leads to a strong dose-dependent lengthening of the circadian period in human U-2 OS and mouse NIH3T3 cells and, most importantly, leads to a significantly altered temperature compensation in U-2 OS. Together with *NUDT21* (also known as *CPSF5* or *CFIm25*), which shows a similar phenotype upon knockdown, *CPSF6* forms a tetramer that binds to UGUA motifs of pre-mRNAs [72], thereby promoting the utilization of distal PASs of its targets [59]. We found that knockdown of *CPSF6* altered 3′ UTR length, mRNA stability, and expression of specific clock genes such as *CLOCK*, *NR1D2*, or *PER3*. However, these properties did not show a systematic temperature dependency (S12 Fig) nor does a CRISPR-Cas9–mediated depletion of *CLOCK* PASs phenocopy the *CPSF6*-knockdown phenotype (see S1 Text), indicating a more complex regulation of temperature decompensation in *CPSF6* knockdown cells. Therefore, we have globally searched for key regulators controlling circadian temperature compensation through the regulation of posttranscriptional processes by extending our analysis to the transcriptome and proteome.

At 37°C, knockdown of *CPSF6* resulted in global shortening of 3′ UTR length in approximately one-third of all detected transcripts, similar to results previously reported for human embryonic kidney cells (HEK-293; [59,73]) and mouse myoblasts (C2C12; [56]). In addition, knockdown of *CPSF6* resulted in global changes in transcript and protein expression, but these can only be attributed to changes in 3′ UTR length to a limited extent, similar to previous observations in T cells that show a major shift towards shorter 3′ UTRs in acivated compared to naive cells. As in our study, this shift in 3′ UTR length is not accompanied by a corresponding global change of mRNA and protein expression levels [67]. How might such *CPSF6*-dependent effects influence circadian temperature compensation? Two conceptually different hypotheses for temperature compensation mechanisms have been proposed. While the balance hypothesis relies on the assumption of counterbalancing the period responses of 2 or

more temperature-dependent reactions, leading to period stability [17,74–78], the critical reaction hypothesis assumes the temperature insensitivity of certain critical period-determining processes [40,45,46,79,80]. Regardless of which hypothesis is correct, we postulated that the temperature response after knockdown of *CPSF6* would be reflected in differential temperature-dependent behavior of either 3′ UTR length, transcript, or protein expression. Indeed, we identified several genes, whose temperature response is *CPSF6* dependent at at least one of the 3 regulatory levels, i.e., 3′ UTR length or transcript and protein abundance, but interestingly, only 6 genes that are regulated by temperature and *CPSF6* at all 3 levels (at our chosen significance thresholds). Among these 6 genes, eukaryotic translation initiation factor 2 subunit 1 (*EIF2S1*, also known as *eIF2Oα*) showed the most significant circadian period phenotype in our large-scale RNAi screen. Interestingly, *EIF2S1* has been described to be a key sensor for a variety of stress conditions such as temperature as well as metabolic fluctuations [81] and has been described to control circadian rhythms in diverse organisms. In mammals, EIF2S1 phosphorylation levels regulate the circadian period by promoting translation of *Atf4* mRNA, a transcription factor targeting binding motifs within the promoters of several clock genes [82]. In addition, clock-controlled rhythmic accumulation of phosphorylated EIF2S1 translationally controls rhythmic gene expression in *Neurospora crassa* [83]. In mouse liver, rhythmic *EIF2S1* expression mediates circadian regulation of stress granula [84]. Being a master regulator of cellular stress response and also a clock regulator, EIF2S1 is a plausible candidate for integrating and processing fluctuating environmental signals to establish a robust circadian period and phase. Future studies are needed to characterize these properties and their dependency on temperature-dependent alternative polyadenylation in detail.

Recently, many transcripts have been shown to exhibit circadian oscillations in 3′ UTR length [85–87]. It would be interesting to find out in future studies whether and to what extent the rhythmicity in 3′ UTR length affects the mechanisms of temperature compensation. In our study, we showed that a small fraction of the transcriptome exhibits temperature-dependent changes in 3′ UTR length. Thus, the previously reported daily rhythms in alternative polyadenylation may be additionally regulated by temperature, potentially allowing the system to plastically shape the circadian transcriptome in response to changing environmental conditions. Our study suggests that alternative polyadenylation is a posttranscriptional mechanism that may underlie temperature compensation of circadian periods in human cells. Alternative polyadenylation has been shown to be highly tissue dependent in diverse organisms such as mammals [50,55,88] or insects [89], very similar to properties of circadian systems [8,90,91]. It would therefore be interesting to investigate in future studies whether alternative polyadenylation acts as an additional regulatory layer to confer tissue-specific fine-tuning of circadian clocks.

Taken together, we here provide evidence that *CPSF6*, a key regulator of PAS selection and, hence, 3′ UTR lengths, modulates not only the period of the mammalian circadian oscillator but also its temperature compensation. By using an integrative omics approach, we found that *CPSF6* affects the transcription and translation of a huge variety of genes in a temperature-dependent manner. Interestingly, in parallel to this study, Kelliher and colleagues [92] reported that knockdown of the *Neurospora* homolog of *CPSF6* significantly alters the compensation of circadian rhythmicity with respect to changes in nutrient levels. These results, in combination with our own data, are in line with the original proposition of Pittendrigh and Caldarola [26] that temperature compensation may "only be a special case of a general homeostatic conservation of the frequency of circadian oscillations in the face of all changes they are likely to encounter in the cell," and *CPSF6* might be a shared molecular constituent. Our data sets and analysis results provide a starting point for future more detailed mechanistic studies.

## 4. Materials and methods

### 4.1. RNAi screen

Large-scale RNAi-based screening for circadian phenotypes has been done as previously described [52,53]. In essence, RNAi constructs have been purchased from Open Biosystems, lentiviruses were produced in HEK293T cells in a 96-well plate format as described earlier [93], and U-2 OS cells harboring a *Bmal1:Luc* reporter construct were transduced with a 100 −$\mu$l virus filtrate plus 8 ng/$\mu$l protamine sulfate. For bioluminescence recordings, U-2 OS cells were maintained in reporter medium containing 500 mL high-glucose DMEM, supplemented with 50 mL FBS, 5 mL 10.000 U/mL penicillin/streptomycin, 500 $\mu$L 10 mg/mL puromycin, and 250 $\mu$M luciferin. Oscillatory properties of bioluminsescence recordings have been estimated using the ChronoStar software, which detrends the raw data by a 24-hour window moving average filter and subsequently fits the detrended data to an exponentially damped cosine function [53]. Statistical properties of circadian oscillatory parameters such as amplitudes, damping coefficients, and periods based on multiple shRNA constructs per gene are assessed via a Redundant SiRNA Activity (RSA) analysis as previously described [94].

In order to further investigate the effect of *EIF2S1* knockdown on the temperature compensation of circadian free-running periods as well as functional interactions with *CPSF6*, we applied an additional small-scale RNAi screen using 2 different bioluminometer setups. For experiments using a TopCount Luminometer, lentiviruses were produced in HEK-293T cells in a 96-well plate as described above. Viral supernatants were filtered, and U-2 OS *Bmal1:Luc* reporter cells were transduced (100 $\mu$L/well + 8 $\mu$g/mL protamine sulfate) with lentiviral shRNA constructs targeting *EIF2S1* (pGIPZ Oligo IDs V2LHS_60800 V2LHS_60804, V2LHS_60802, V2LHS_5357) or with nonsilencing control in a 96-well format. The next day, viral supernatant was exchanged for puromycin-containing (10 $\mu$g/mL) medium and cells were incubated for 2 days. Following, cells were synchronized with 1 $\mu$M dexamethasone for 30 to 45 minutes before cells were supplemented with 200 $\mu$L/well reporter medium and bioluminescence was recorded for approximately 7 days in a TopCount luminometer with a stacker unit at a sampling rate of 30 minutes at 3 different temperatures, 32°C, 35°C, or 39°C. For experiments using LumiBoxes, lentiviruses were produced in HEK-293T cells in T-75 flasks by CalPhos transfection according to the manufacturer's instructions. Viral supernatants were filtered and U-2 OS *Bmal1:Luc* reporter cells were transduced (1.5 mL/dish + 8 $\mu$g/mL protamine sulfate) with lentiviral shRNA constructs targeting *EIF2S1* (pGIPZ Oligo ID V2LHS_60800) or nonsilencing control in a 35-mm dish format. The next day, viral supernatant was exchanged for puromycin-containing (10 $\mu$g/mL) medium and cells were incubated for 2 days. Following, cells were synchronized with 1 $\mu$M dexamethasone for 30 to 45 minutes before cells were supplemented with 2 mL/dish reporter medium and bioluminescence was recorded for approximately 7 days in light-tight single photomultiplier tubes ("LumiBoxes") at 32°C, 35°C, and 39°C. For both experimental setups, data have been analyzed by the ChronoStar software as described above. Subsequently, results of individual screens (even across different shRNA constructs) have been averaged; see also S12 Table.

In order to increase the confidence in phentopyes found within our RNAi screens, generally multiple individual shRNAs have been used to target a single given gene as suggested, e.g., by [95]. Corresponding nucleotide sequences or pGIPZ Oligo IDs can be found in S1 and S12 Tables, respectively.

### 4.2. Statistical accession of differential temperature responses

To statistically access whether a certain observable of interest (i.e., the free-running period $\tau$, 3′ UTR length, and transcript or protein abundance) exhibits different responses upon adiabatic

changes in environmental temperature between wild-type and *CPSF6* knockdown cells, we follow a method as outlined in Chapter 11.4 of [96]. In essence, we assume (for the sake of simplicity) that the dependency of the observable of interest upon temperature can be described by a linear relationship for each of the 2 groups (wild type and *CPSF6* knockdown) and test against the null hypothesis that the slopes $b_i$ ($i$ = 1,2) describing such dependency are identical; see Chapter 11.4 of [96] for details and S2 Fig for examples. In case of the transcriptomics and proteomics data analysis, *P* values are corrected for multiple testing via the Benjamini–Hochberg procedure.

## 4.3. Sample collection

U-2 OS BLH cells were transduced with RNAi lentiviruses with bulk method. In brief, U-2 OS cells grown in 10 T175 flasks were taken in suspension by trypsin digest and evenly distributed to eight 50 mL Falcon tubes. Pelleted cells were carefully resuspended in 4.5 mL lentiviral supernatant (either pGIPZ nonsilencing or pGIPZ targeting *CPSF6*, V3LHS_640891) containing protamine sulfate (at 8 $\mu$g/mL final concentration). Cells are spun at 37˚C, 800 × *g* for 90 minutes. After centrifugation, cells are gently resuspended using a 1-mL pipette and seeded to 3 × T75 flasks per 50 mL Falcon tube (in total 24 × T75 flasks) and incubated at 37˚C. Cells were transfered to respecitve temperature conditions (32˚C, 37˚C or 39˚C) 5 days after transduction. Cells were harvested for RNA or protein isolation 4 days after transfer to respective temperature conditions.

## 4.4. Protein extraction

Subsequent procedures have been performed on ice or at 4˚C. Cells grown in cell culture flasks (75 cm$^2$) were lysed by adding 0.5 ml of lysis buffer containing 4% SDS in 100 mM Tris (pH 8.5). Lysates were collected with cell scraper, transfered to Eppendorft tubes, boiled for 15 minutes at 95˚C, and stored at −80˚C.

## 4.5. RNA-sequencing

Two protocols have been applied to pursue different goals: To precisely map PASs in wild-type U-2 OS cells, we applied the 3′Seq RNAseH$^+$ protocol as previously described [63]. To quantify overall transcript expression and 3′ UTR length (NCOM) in wild-type and *CPSF6* knockdown cells, we applied 3′ end seq without RNAseH digest (RNAseH$^-$) to preserve relative quantities of RNA fragments.

A schematic drawing of the analysis pipeline can be found in S13 Fig.

**4.5.1. RNAseH$^+$ protocol.** U-2 OS wild-type and *CPSF6* knockdown cells are cultured at 37˚C. After 4 days, cells are harvested and RNA extracted using a standard isolation kit (Invitrogen Pure Link RNA Mini kit), Zn-fragmented and fragments with poly(A) tails are selected using oligo-(dT) beads. These preselected fragments are subsequently digested by an RNAse H treatment (RNAseH$^+$-seq), which removes the poly(A)-RNA and allows for an exact linker ligation at the cleavage and polyadenylation site [63,97]. Strand-specific reads of this 3′ end sequence library are aligned to the reference genome GRCh37 (hg19) and expression is quantified by the End Sequence Analysis Toolkit (ESAT; [97]). Polyadenylation signals are searched up to 5,000 bp beyond the annotated end of transcripts.

**4.5.2. RNAseH$^-$ protocol.** In case of newly identified PASs through our RNAseH$^+$ protocol, we extended the reference genome annotation by the newly identified PAS. Extended isoforms are marked by "_ext" to the RefSeqID in S3, S4, S6, S7, S9, and S10 Tables. This new annotation has then been used for a further 3′ UTR expression analysis. 3′ end identification and expression quantification has been again done via ESAT, using a scanning window length

of 50 bases. Reads are subsequently normalized using DESeq [98], and transcript isoforms with identical 3′ UTRs are merged together.

## 4.6. Proteomics

Four independent replicates of U-2 OS cell cultures at 3 temperature conditions for wild-type and *CPSF6* knockdown cells were subjected to quantitative MS-based proteomics analysis. In brief, frozen cell lysates in 4% SDS buffer were prepared for MS-based quantitative proteomics analysis. Briefly, samples were boiled for 5 minutes at 95°C, and homogenates were sonicated using a Bioruptor (2×15 cycles at 4°C; 30 seconds ON, 30 seconds OFF each cycle, at maximum power). Once a homogeneous solution was formed, the protein concentration was estimated based on Tryptophan assay. About 1 mg of protein from each sample was used as starting material for protein digestion. For this, protein lysates were incubated for 20 minutes at RT with 2.5 mM Dithiothreitol (DTT) and 27.5 mM 2-Chloroacetamide (CAA) for protein reduction and alkylation, respectively. The lysates were then precipitated with 80% acetone (overnight incubation at −20°C) and collected the following day by centrifugation ($1,500 \times g$, 10 minutes, 4°C). After the protein pellets were washed 4 times with ice-cold 80% acetone and air-dried at RT for 20 minutes, 500 $\mu$L of trifluoroethanol (TFE) solution was added to each sample. Pellets were resuspended by sonication using a Bioruptor (15 cycles at 4°C 30 seconds ON, 30 seconds OFF each cycle, at maximum power) and subsequently incubated with endopeptidase LysC for 1 hour at RT followed by overnight incubation with Trypsin at 37°C. For both digesting enzymes, the proportion to the protein amount was 1:100. After digestion, the samples were centrifuged ($10,000 \times g$, 10 minutes, RT) to remove potential cellular debris, and 20 $\mu$g were used for peptide desalting and clean-up using SDB-RPS StageTips. After eluting (80% ACN, 2.5% NH4OH), the peptides were concentrated in a centrifugal evaporator (45°C) until dryness and were reconstituted in LC-MS loading buffer (2% ACN, 0.2% TFA) and the eluate was stored at −20°C. A volume corresponding to 400 ng of peptides was used for the analysis using an LC 1200 ultra-high-pressure system (Thermo Fisher Scientific) coupled via a nano-electrospray ion source (Thermo Fisher Scientific) to a Q Exactive HF-X Orbitrap (Thermo Fisher Scientific). Prior to MS, the peptides were separated on a 50-cm reversed-phase column (diameter of 75 mm packed in-house with ReproSil-Pur C18-AQ 1.9 mm resin [Dr. Maisch GmbH]) over a 120-min gradient of 5% to 60% buffer B (0.1% formic acid and 80% ACN). Full MS scans were acquired in the 300 to 1,650 m/z range (R = 60,000 at 200 m/z) at a target of 3e6 ions. The 15 most intense ions were isolated, fragmented with higher-energy collisional dissociation (HCD) (target 1e5 ions, maximum injection time 120 ms, isolation window 1.4 m/z, NCE 27%, and underfill ratio of 20%), and finally detected in the Orbitrap (R = 15,000 at 200 m/z). Raw MS data were processed with MaxQuant [99], and reported LFQ normalized protein intensities were used for further bioinformatic analysis.

## Supporting information

**S1 Fig. Circadian free-running period dose dependently lengthens upon *CPSF6* and *NUDT21* knockdown.** (**A**) Raw (first column) and detrended (second column) *Bmal1-luciferase* expression in human control (blue) and *CPSF6*-depleted (red) U-2 OS cells, using 3 different shRNA constructs. Corresponding periods and relative amplitudes as determined by the ChronoStar software are depicted in the third and fourth columns, respectively. Comparing the relative *CPSF6* mRNA expression (**B**) with the corresponding periods (**A**, third column) reveals a dose-dependent (i.e., knockdown efficiency–dependent) lengthening of the free-running period in U-2 OS upon shRNA-mediated *CPSF6* depletion. (**C**) Raw (first column) and detrended (second column) *Bmal1-luciferase* expression in mouse control (blue) and *CPSF6*-

depleted (red) NIH3T3 cells, using 3 different shRNA constructs, together with the corresponding periods (third column) and relative amplitudes (fourth column). A comparison between the relative *CPSF6* mRNA expression (**D**) and the corresponding period (**C**, third column) reveals a dose-dependent lengthening upon shRNA-mediated *CPSF6* depletion in mouse NIH3T3 cells. (**E**) Raw (first column) and detrended (second column) *Bmal1-luciferase* expression in human control (blue) and *NUDT21* (also known as *CPSF5*) depleted (red) U-2 OS cells, using 3 different shRNA constructs, together with the corresponding periods (third column) and relative amplitudes (fourth column). A comparison between the relative *NUDT21* mRNA expression (**F**) and the corresponding period (**E**, third column) reveals a dose-dependent lengthening upon shRNA-mediated *NUDT21* depletion in human U-2 OS cells. Raw data underlying panels A-F can be found in S2 and S5 Data.
(TIFF)

**S2 Fig. Knockdown of *CPSF6* or *NUDT21* leads to an altered temperature compensation of the circadian period in U-2 OS.** (**A**) Temperature response of circadian free-running period $\tau$ is statistically different ($p<0.001$) in wild-type and *CPSF6* knockdown cells, assuming a linear dependency of free-running period $\tau$ on temperature $T$ and testing against the null hypothesis that the 2 slopes, $b_1 \approx -0.13 \frac{h}{°C}$ and $b_2 \approx -0.45 \frac{h}{°C}$, for wild-type ($n_1 = 16$) and *CPSF6* knockdown ($n_2 = 16$) cells, respectively, are identical; see Materials and methods. Temperature coefficients have been obtained, using the equation $Q_{10} = \left(\frac{\tau_1}{\tau_2}\right)^{10°C/7°C}$ with $\tau_1$ and $\tau_2$ being the free-running period determined at 32°C and 39°C as obtained from the linear regression, respectively. Error propagation has been calculated via the Python uncertainties package. (**B**) Similarly, the temperature response of circadian free-running period $\tau$ is statistically significantly different ($p≈0.001$) in control ($n_1 = 7$) and *NUDT21* ($n_2 =8$) depleted cells. Raw data and code underlying panels A and B can be found in S3 Dataset.
(TIFF)

**S3 Fig. *CPSF6* knockdown induced changes in polyadenylation site usage, mRNA expression, and mRNA half-life among canonical clock genes.** (**A**) Dynamics of *CPSF6*, *ARNTL*, *CRY1*, *CRY2*, *NR1D1*, *PER1*, and *PER2* mRNA expression after application of the transcription inhibitor triptolide at time point 0 hours in wild-type (blue) and *CPSF6*-depleted (red) cells. Half-lives of the corresponding mRNAs are determined from the linear fits to the logarithmic data as depicted by the straight lines. (**B**) Bar plots, comparing the mRNA half-lives (left), total mRNA expression (middle), and distal polyadenylation site usage (right) in wild-type (blue) versus *CPSF6*-depleted cells. Each dot denotes results from an individual experiment where values of 3 technical replicates have been averaged. Bars denote the corresponding averages of these individual experiments. Results in (**A**) show dynamics of 3 technical replicates within 1 individual experiment in wild-type or *CPSF6* knockdown cells, thus corresponding to 1 dot in the bottom panel. *P* values for a statistical comparison (*t* test) between the total mRNA half-life, total mRNA expression, and distal PAS usage between control and *CPSF6*-depleted cells are $p_a = 0.819$, $p_b = 0.296$, $p_c = 0.191$, $p_d = 0.520$, $p_e = 0.863$, $p_f = 0.754$, $p_g = 0.238$, $p_h = 0.442$, $p_i = 0.196$, $p_j = 0.523$, $p_k = 0.424$, $p_l = 0.021$, $p_m = 0.230$, $p_n = 0.997$. (**C**) Schematic representation of primer location, used to measure expression of total- (dark green) and long-3′ UTR (light green) isoform expression of the 9 core clock genes studied in Figs 1E, 1F, S3A and S3B. The distal polyadenylation site (PAS) usage is defined by the ratio of the long-3′ UTR expression (light green) divided by the total expression (dark green). While light gray regions depict the coding strand, dark gray regions depict 5′ and 3′ UTRs. NCBI Reference Sequence IDs, together with the corresponding length in base pairs (bp) and the gene symbol are given next to the schematic representation. Raw data underlying panels A and B can be found in S4 and

S5 Data.
(TIFF)

**S4 Fig. Number of polyadenylation sites per gene.** The RNAseH⁻ approach detects alternative polyadenylation signals for more than 70% of the genes. Raw data underlying this figure can be found in S2 Table.
(TIFF)

**S5 Fig. Depletion of *CPSF6* induces a global shift towards shorter 3′ UTRs.** (**A**) Histogram of the center of read distribution along the 3′ UTR, normalized to the length of the isoform's 3′ UTR (NCOM), in control (blue) and *CPSF6*-depleted (red) cells. Here, the distribution only considers those isoforms that show a significant shift in their NCOM upon *CPSF6* knockdown. (**B**) Scatter plot of NCOM values in control versus *CPSF6*-depleted cells. Orange and gray dots depict isoforms exhibiting a significant or no significant shift in their NCOM value upon *CPSF6* knockdown, respectively. (**C**) Histogram of the nonnormalized center of read distribution along the isoforms 3′ UTR (COM) in control (blue) and *CPSF6*-depleted (red) cells. (**D**) Decadic logarithm of COM values in control versus *CPSF6*-depleted cells. Again, orange and gray dots depict isoforms exhibiting a significant or no significant shift in their COM value upon *CPSF6* knockdown, respectively. As in Fig 2B and 2C of the main text, all data are from experiments at 37˚. Data underlying this figure are available from the NCBI Gene Expression Omnibus (GEO; http://www.ncbi.nlm.nih.gov/geo/) under accession number GSE185896 and is contained in S3 Table.
(TIFF)

**S6 Fig. Quality control of proteomics data.** (**A**) Principal component analysis on log2-transformed protein LFQ intensities as determined by the MaxQuant software [99]. Technical replicates as indicated by equal coloring group together. (**B**) Correlogram of the Pearson correlation coefficients (r) from the log2-transformed protein LFQ intensities across measured samples. As expected, replicates with the same genetic background (wild-type versus *CPSF6* knockdown) and at the same environmental temperature show the highest correlations of abundance values. (**C**) Representative scatter plots of log2 transformed protein LFQ intensities across different samples with the Pearson correlation coefficient (*r*). Data underlying this figure are available from the ProteomeXchange Consortium via the PRIDE partner repository (http://www.ebi.ac.uk/pride/archive/) with the dataset identifier PXD029343.
(TIFF)

**S7 Fig. Genes with most significant variations in 3′ UTR length, transcript expression, and protein abundance upon changes in environmental temperature.** (**A**) Heatmap of 25 NCOM values that exhibit the most significant shortening (upper panel) or lengthening (bottom panel) of 3′ UTRs upon changes in environmental temperature in wild-type U-2 OS cells, sorted by increasing Benjamini–Hochberg corrected *P* values. All replicates at all 3 temperatures, namely 32˚C, 37˚C, and 39˚C, are shown. (**B**) Same as panel (**A**), showing the 25 genes with the most significant down- (upper panel) or up-regulation (bottom) in gene expression upon increasing environmental temperatures. (**C**) Same as panels (**A**) and (**B**), showing the 25 genes with the most significant down- (upper panel) or up-regulation (bottom) in protein abundance upon increasing environmental temperatures. Data underlying this figure are available from the NCBI Gene Expression Omnibus (GEO; http://www.ncbi.nlm.nih.gov/geo/) under accession number GSE185896 and the ProteomeXchange Consortium via the PRIDE partner repository (http://www.ebi.ac.uk/pride/archive/) with the dataset identifier PXD029343 and are contained in S6–S8 Tables.
(TIFF)

**S8 Fig. Changes in 3′ UTR length, transcript abundance, and protein abundance upon temperature changes in *CPSF6* knockdown cells.** (**A**) MA plot showing the shift in 3′ UTR length upon temperature variations between 32˚C and 39˚C in *CPSF6*-depleted cells. Transcripts with a statistically significant change in 3′ UTR length at a 5% FDR (Benjamini–Hochberg corrected *P* values) are depicted by orange markers. No global shift in 3′ UTR length can be observed upon temperature alterations in *CPSF6*-depleted cells (inset). (**B**) Same as panel (**A**), showing the binary logarithm of transcript expression fold changes upon temperature variations between 32˚C and 39˚C on the ordinate. Transcripts with a statistically significant change in expression at a 5% FDR are depicted by green markers. (**C**) Same as panel (**B**), showing the binary logarithm of protein LFQ intensity fold changes upon temperature variations between 32˚C and 39˚C on the ordinate as well as the binary logarithm of the mean among all temperatures from the corresponding proteins. Proteins with a statistically significant change in abundance at a 5% FDR are depicted by pink markers. (**D**) Heatmap of 25 NCOM values that exhibit the most significant shortening (upper panel) or lengthening (bottom panel) of 3′ UTRs upon changes in environmental temperature in *CPSF6*-depleted U-2 OS cells, sorted by increasing Benjamini–Hochberg corrected *P* values. All replicates at all 3 temperatures, namely 32˚C, 37˚C and 39˚C, are shown. (**E**) Same as panel (**D**), showing the 25 genes with the most significant down- (upper panel) or up-regulation (bottom) in gene expression upon increasing environmental temperatures. (**F**) Same as panels (**D**) and (**E**), showing the 25 genes with the most significant down- (upper panel) or up-regulation (bottom) in protein abundance upon increasing environmental temperatures. Data underlying this figure are available from the NCBI Gene Expression Omnibus (GEO; http://www.ncbi.nlm.nih.gov/geo/) under accession number GSE185896 and the ProteomeXchange Consortium via the PRIDE partner repository (http://www.ebi.ac.uk/pride/archive/) with the dataset identifier PXD029343.
(TIFF)

**S9 Fig. Read distribution for genes exhibiting differential temperature responses between wild-type and *CPSF6* knockdown cells.** 3′ UTR read distributions (bars) and corresponding NCOM mean values (vertical bold lines) with standard deviations (shaded areas) of all 6 genes that show a differential temperature response at the level of 3′ UTR length, transcript expression, as well as protein abundance, shown for wild type (blue) and *CPSF6* knockdown cells (red) at 3 different temperatures. The *n* = 3 technical replicates per condition are overdubbed in the bar plots. Data underlying this figure are available from the NCBI Gene Expression Omnibus (GEO; http://www.ncbi.nlm.nih.gov/geo/) under accession number GSE185896.
(TIFF)

**S10 Fig. Large-scale RNAi screen example time traces.** Relative amplitude of *Bmal1-luciferase* oscillations for RNAi constructs (black) and the corresponding control (plate mean; gray). Depicted are examples for genes showing a significant period shortening or lengthening in the RNAi screen as well as a differential temperature response at the NCOM, transript, and protein level (first row), the NCOM and transcript level (second and third row), the NCOM and protein level (fourth and fifth row), or the transcript and protein level (sixth and seventh row); compare Fig 5D.
(TIFF)

**S11 Fig. *EIF2S1* knockdown diminishes temperature compensation.** (**A**) Oscillatory periods of wild-type (blue) and *EIF2S1* knockdown cells (red) are determined at 3 different temperatures as described in Materials and methods. While differences between control and *EIF2S1*-depleted cells are significantly different at 35˚C and 39˚C ($p < 0.01$, *t* test), differences are

slightly nonsignificant at 32°C ($p \approx 0.068$, $t$ test) even though showing the same systematic period lengthening in knockdown cells. (**B**) As for the period difference at 32°C, even though linear relationships describing the temperature response of the circadian free-running period are not significantly different in control and *EIF2S1*-depleted cells ($p \leq 0.2$), these data are consistent with the altered temperature compensation phenotype observed in *CPSF6*-depleted cells. Here, straight lines denote linear regressions as used to determine the corresponding period coefficient ($Q_{10}$). Circle markers denote experiments exerted by using a TopCount luminometer, while triangle markers denote experiments exerted in light-tight single photo-multiplier tubes (LumiBoxes). Data underlying this figure are available from S12 Table.
(TIFF)

**S12 Fig. Canonical clock genes do not show a systematic temperature dependency in NCOM values and transcript abundance.** 3′ UTR read distributions (bars) and corresponding NCOM mean values (vertical bold lines) with standard deviations (shaded areas) of all 9 canonical clock genes investigated in Figs 1 and S3 for wild-type (blue) and *CPSF6* knockdown cells (red) at 3 different temperatures. The $n = 3$ technical replicates per condition are over-dubbed in the bar plots. Data underlying this figure are available from the NCBI Gene Expression Omnibus (GEO; http://www.ncbi.nlm.nih.gov/geo/) under accession number GSE185896.
(TIFF)

**S13 Fig. Schematic drawing of the transcriptomics analysis pipeline.**
(TIFF)

**S1 Table. Summary of RNAi screen results.** Within our large-scale RNAi screen, a single gene is often targeted by multiple shRNA constructs. Columns H and I depict the number of constructs and corresponding nucleotide sequences, respectively. Gene symbols of the shRNA targets are given in column A. Target oscillatory properties of *Bmal1-luciferase* expression are estimated using the ChronoStar software; see Materials and methods. Using the RSA method, we statistically access whether an oscillatory property such as the amplitude, damping rate, or period significantly deviates from the norm in one or the other direction, i.e., 2 *P* values per parameter are given. Columns B-G depict decadic logarithms of the corresponding *P* values.
(XLS)

**S2 Table. Polyadenylation sites detected by the RNAseH-seq approach.** Columns A-F depict the chromosome information, the start and end position of the polyadenylation site, the corresponding gene symbol, ESAT score, and strand information, respectively, of the polyadenylation site found by our RNAseH-seq approach.
(XLSX)

**S3 Table. Isoforms showing a significant shift in 3′ UTR length (NCOM) upon CPSF6 knockdown at 37°C.** All 4,060 isoforms showing a significant shift in NCOM values upon *CPSF6* knockdown at 37°C are deposited in the table. Column A shows the RefSeq IDs associated with a given isoform. Since we focus on different 3′ UTR isoforms, sometimes more than 1 RefSeq ID can be associated with a given isoform. RefSeq IDs that have been reannotated, using the newly found polyadenylation sites from S2 Table, have the extension "_ext." Columns B, C, and D contain the corresponding gene symbol as well as chromosome and strand information of the isoform, respectively. Columns E-J contain the NCOM values for each of the 3 technical replicates within control (WT) and *CPSF6* (KD) depleted cells. Columns K and L contain the *P* values as well as Benjamini–Hochberg corrected *P* values of the binary

comparison (*t* test) of NCOM values between control and *CPSF6* knockdown cells, respectively.
(XLS)

**S4 Table. Isoforms showing a significant differential transcript expression between wild-type and CPSF6 knockdown cells at 37˚C.** All 1,924 differentially expressed isoforms between wild-type and *CPSF6*-depleted cells at 37˚C as detected by the nbinomTest function of DESeq are deposited in the table. Columns A-D depict the RefSeq IDs, gene symbols, as well as chromosome and strand information of the corresponding isoforms. Columns E-J contain DGE values as determined by ESAT for all 3 technical replicates of control (WT) and *CPSF6* (KD)-depleted cells. Columns K and L depict the *P* values and Benjamini–Hochberg corrected *P* values as determined by DESeq for rejecting the null hypothesis that the mean DGE values in control and *CPSF6*-depleted cells are equal.
(XLS)

**S5 Table. Proteins showing a significantly altered abundance in wild-type and CPSF6 knockdown cells at 37˚C.** Information about all 1,683 peptides showing a significantly different abundance after *CPSF6* knockdown at 37˚C is deposited in the table. Column A depicts the corresponding gene symbol. Columns B-I contain the LFQ intensities for all 4 technical replicates of control (WT) and *CPSF6* (KD)-depleted cells. Column J depicts the binary logarithms of the fold changes between control and *CPSF6*-depleted cells. Columns K and L contain the *P* values as well as Benjamini–Hochberg corrected *P* values of the binary comparison (*t* test) of LFQ intensities (i.e., peptide abundances) between control and *CPSF6* knockdown cells, respectively.
(XLS)

**S6 Table. Isoforms showing a significant shift in 3′ UTR length upon temperature changes in wild-type cells.** Information about all 244 isoforms showing a significant shift in NCOM values upon temperature changes from 32˚C to 39˚C in wild-type cells is deposited in the table. Columns A-D depict the RefSeq IDs, gene symbols, as well as chromosome and strand information of the corresponding isoforms. Columns E-M contain the NCOM values for each of the 3 technical replicates in control (WT) cells at all 3 tested temperatures, namely 32˚C, 37˚C, and 39˚C. Columns N-Q depict the slope and intercept values as well as corresponding *P* values and Benjamini–Hochberg corrected *P* values, respectively, of the linear regression, using temperature as independent and NCOM values as dependent variables.
(XLS)

**S7 Table. Isoforms showing a significant alteration in transcript expression upon temperature changes in wild-type cells.** Information about all 2,445 isoforms showing a significant alteration of DGE values (i.e., transcript abundance) upon temperature changes from 32˚C to 39˚C in wild-type cells is deposited in the table. Columns A-D depict the RefSeq IDs, gene symbols, as well as chromosome and strand information of the corresponding isoforms. Columns E-M contain the DGE values for each of the 3 technical replicates in control (WT) cells at all 3 tested temperatures, namely 32˚C, 37˚C, and 39˚C. Columns N-Q depict the slope and intercept values as well as corresponding *P* values and Benjamini–Hochberg corrected *P* values, respectively, of the linear regression, using temperature as independent and DGE values as dependent variables.
(XLS)

**S8 Table. Proteins showing a significantly altered abundance upon temperature changes in wild-type cells.** Information about all 1,965 peptides showing a significant alteration of LFQ

intensities (i.e., peptide abundance) upon temperature changes from 32˚C to 39˚C in wild-type cells is deposited in the table. Column A depicts the corresponding gene symbols. Columns B-M contain the LFQ intensities for each of the 4 technical replicates in control (WT) cells at all 3 tested temperatures, namely 32˚C, 37˚C, and 39˚C. Columns N-Q depict the slope and intercept values as well as corresponding *P* values and Benjamini–Hochberg corrected *P* values, respectively, of the linear regression, using temperature as independent and LFQ intensities as dependent variables.
(XLS)

**S9 Table. Isoforms showing a significantly differential response of 3′ UTR length between wild-type and *CPSF6* knockdown cells upon temperature changes.** Information about all 191 isoforms showing a significantly differential temperature response in NCOM values (i.e., 3′ UTR length) between wild-type (WT) and *CPSF6*-depleted (KD) cells. Columns A-D depict the RefSeq IDs, gene symbols, as well as chromosome and strand information of the corresponding isoforms. Columns E-V contain the NCOM values for each of the 3 technical replicates in control (WT) and *CPSF6*-depleted cells at all 3 tested temperatures, namely 32˚C, 37˚C, and 39˚C. Columns W-Z depict the slope and intercept values of the linear regressions in wild-type and *CPSF6*-depleted cells, using temperature as independent and NCOM values as dependent variables. Columns AA and AB depict corresponding *P* values and Benjamini–Hochberg corrected *P* values, respectively, obtained from our statistical test against the null hypothesis that the slopes describing the temperature dependency are identical in wild-type and *CPSF6*-depleted cells; see Materials and methods for details.
(XLS)

**S10 Table. Isoforms showing a significantly differential response of transcript expression between wild-type and *CPSF6* knockdown cells upon temperature changes.** Information about all 994 isoforms showing a significantly differential temperature response in DGE values (i.e., transcript abundance) between wild-type (WT) and *CPSF6*-depleted (KD) cells. Columns A-D depict the RefSeq IDs, gene symbols, as well as chromosome and strand information of the corresponding isoforms. Columns E-V contain the DGE values for each of the 3 technical replicates in control (WT) and *CPSF6*-depleted cells at all 3 tested temperatures, namely 32˚C, 37˚C, and 39˚C. Columns W-Z depict the slope and intercept values of the linear regressions in wild-type and *CPSF6*-depleted cells, using temperature as independent and DGE values as dependent variables. Columns AA and AB depict corresponding *P* values and Benjamini–Hochberg corrected *P* values, respectively, obtained from our statistical test against the null hypothesis that the slopes describing the temperature dependency are identical in wild-type and *CPSF6*-depleted cells; see Materials and methods for details.
(XLS)

**S11 Table. Proteins showing a significantly differential response of abundance between wild-type and *CPSF6* knockdown cells upon temperature changes.** Information about all 678 peptides showing a significantly differential temperature response in LFQ intensities (i.e., peptide abundance) between wild-type (WT) and *CPSF6*-depleted (KD) cells. Column A depicts the corresponding gene symbols. Columns B-Y contain the LFQ intensities for each of the 4 technical replicates in control (WT) and *CPSF6*-depleted cells at all 3 tested temperatures, namely 32˚C, 37˚C, and 39˚C. Columns Z-AC depict the slope and intercept values of the linear regressions in wild-type and *CPSF6*-depleted cells, using temperature as independent and LFQ intensities as dependent variables. Columns AD and AE depict corresponding *P* values and Benjamini–Hochberg corrected *P* values, respectively, obtained from our statistical test against the null hypothesis that the slopes describing the temperature dependency are identical

in wild-type and *CPSF6*-depleted cells; see Materials and methods for details.
(XLS)

**S12 Table. Circadian parameters obtained from *EIF2S1* temperature compensation RNAi screen.** Experiments have been performed as described in Materials and methods. Raw bioluminescence recordings are detrended by a 24-hour moving average and analyzed via the *ChronoStar* software. Results of the nonlinear fitting by *ChronoStar* are depicted in the last 6 columns of each data block. Data where the bioluminescence after transduction and selection of U-2 OS cells is low compared to average (red font) as well as those where the error of the nonlinear curve fit is larger than 0.3 (red shaded cells) are sorted out for the further analysis in S11 Fig. Note that sheet 1 contains results for shRNA constructs targeting *EIF2S1*, while sheet 2 contains results for the nonsilencing constructs.
(XLSX)

**S1 Text. CRISPR-Cas9–mediated depletion of *CLOCK* 3′ UTR polyadenylation sites does not phenocopy the *CPSF6* knockdown period lengthening phenotype.**
(PDF)

**S1 Data. Data underlying Fig 1.** Separate csv files containing the data underlying subpanels A, B, C, E, and F of Fig 1 of the main text.
(ZIP)

**S2 Data. Data underlying S1 Fig.** Separate csv files containing the data underlying all subpanels of S1 Fig.
(ZIP)

**S3 Data. Data underlying S2 Fig.** Python code containing the data and statistical test for differential slopes underlying subpanels A and B of S2 Fig.
(PY)

**S4 Data. Data underlying S3 Fig.** Separate csv files containing the data underlying subpanels A and B of S3 Fig.
(ZIP)

**S5 Data. Data underlying Figs 1, S1 and S3.** Graph pad prism file containing data, statistics, and visualizations underlying panels A, B, C, E, and F of Fig 1; all panels of S1 Fig and panels A and B of S3 Fig.
(PZFX)

**S1 Raw Images. Uncropped gel related to Supplementary text figure 2 of S1 Text together with the corresponding annotation.**
(PDF)

## Acknowledgments

We gratefully acknowledge Astrid Grudziecki and Maike Mette Thaben for excellent technical assistance as well as Paul Thaben for support in statistical analysis.

## Author Contributions

**Conceptualization:** Christoph Schmal, Bert Maier, Achim Kramer.

**Data curation:** Christoph Schmal, Bert Maier.

**Formal analysis:** Christoph Schmal, Bert Maier, Reut Ashwal-Fluss, Osnat Bartok, Anna-Marie Finger, Tanja Bange, Stella Koutsouli, Maria S. Robles, Sebastian Kadener, Hanspeter Herzel, Achim Kramer.

**Funding acquisition:** Christoph Schmal, Maria S. Robles, Sebastian Kadener, Hanspeter Herzel, Achim Kramer.

**Investigation:** Christoph Schmal, Bert Maier, Anna-Marie Finger, Tanja Bange, Stella Koutsouli, Maria S. Robles, Sebastian Kadener, Hanspeter Herzel, Achim Kramer.

**Methodology:** Christoph Schmal, Bert Maier, Maria S. Robles.

**Project administration:** Achim Kramer.

**Resources:** Maria S. Robles, Achim Kramer.

**Software:** Bert Maier, Reut Ashwal-Fluss, Osnat Bartok, Sebastian Kadener.

**Supervision:** Achim Kramer.

**Visualization:** Christoph Schmal, Bert Maier.

**Writing – original draft:** Christoph Schmal, Bert Maier, Hanspeter Herzel, Achim Kramer.

**Writing – review & editing:** Christoph Schmal, Bert Maier, Maria S. Robles, Sebastian Kadener, Hanspeter Herzel, Achim Kramer.

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
