## [Editor Report · Decision Letter 0]

27 Jun 2022

Dear Dr Schmal, 

Thank you for submitting your manuscript entitled "Alternative polyadenylation factor CPSF6 regulates temperature compensation of the mammalian circadian clock" for consideration as a Methods and Resources by PLOS Biology.

Your manuscript has now been evaluated by the PLOS Biology editorial staff as well as by an academic editor with relevant expertise and I am writing to let you know that we would like to send your submission out for external peer review.

Once your full submission is complete, your paper will undergo a series of checks in preparation for peer review. After your manuscript has passed the checks it will be sent out for review. To provide the metadata for your submission, please Login to Editorial Manager (https://www.editorialmanager.com/pbiology) within two working days, i.e. by Jun 29 2022 11:59PM.

As a note, after discussion within the team, we think that your manuscript would be best suited for our Research Article format (https://journals.plos.org/plosbiology/s/what-we-publish#loc-research-articles). Therefore, when resubmitting, we ask that you change the article type, accordingly. This should not involve any formatting changes/etc, but do let me know if you have any questions or concerns about this request. 

Kind regards,

Luke

Lucas Smith, Ph.D.

Associate Editor

PLOS Biology

lsmith@plos.org

---

## [Decision Letter · Decision Letter 1]

25 Aug 2022

Dear Dr Schmal,

Thank you again for your patience while your manuscript "Alternative polyadenylation factor CPSF6 regulates temperature compensation of the mammalian circadian clock" was peer-reviewed at PLOS Biology, and please accept our apologies for the delay in sending you a decision. Your manuscript has now been evaluated by the PLOS Biology editors, an Academic Editor with relevant expertise, and by several independent reviewers.

The reviews are appended below. As you will see below, while the reviewers have commented that the data is of high quality and that the main conclusion of the study is interesting, they have also raised a number of important concerns which would need to be addressed before we could consider your manuscript for publication at PLOS Biology. The reviewers have highlighted the need for additional validation and identified potential issues with the experimental design and data interpretations which undermine the strength of the conclusions. Moreover, Reviewer 2 has commented that the study lacks deep mechanistic insights.

After careful discussion with the Academic Editor, we do not feel able to accept the manuscript in its current form, but would be willing to consider a revised version of your manuscript which provides stronger mechanistic insights and thoroughly addresses the reviewer concerns, with new data as needed.

Given the extent of revision needed, we cannot make a decision about publication until we have seen the revised manuscript and your response to the reviewers' comments. Your revised manuscript is likely to be sent for further evaluation by all or a subset of the reviewers.

We expect to receive your revised manuscript within 3 months, however we realize we are asking a lot in the revision and so would be happy to grant an extension if needed. Please email us (plosbiology@plos.org) if you have any questions or concerns, or would like to request an extension

**IMPORTANT - SUBMITTING YOUR REVISION**

*Re-submission Checklist*

*Published Peer Review*

*PLOS Data Policy*

*Blot and Gel Data Policy*

Sincerely,

Lucas

Lucas Smith, Ph.D.

Associate Editor

PLOS Biology

lsmith@plos.org

REVIEWS:

Reviewer #1: In this manuscript, Schmal et al., examined the role of CPSF6 in regulating circadian temperature compensation, one of the defining characteristics of circadian rhythms. The molecular mechanisms underlying temperature compensation are indeed poorly understood and more studies in this area are definitely needed. The study includes a large quantity of data to shed light into this, however, the data presented in this manuscript do not support the authors' major conclusion and lack novelty. Some of the experimental design or data interpretation are not rational to this reviewer either. Please see the comments below for details. Because of these reasons, the reviewer cannot support the publication of this study. 

Major points:

* The authors concluded that CPSF6 regulates temperature compensation based on Figure 1B, in which they knocked down CPSF6 and measured the period of Bmal1-luc in U2OS cells at three different temperatures. First, the authors need to test whether the efficiency of knock-down (i.e., the activities of Dicer, Argonaute, and RISC complex) and 3'-end processing are affected by the difference in temperatures to support their conclusion.

* Second: Q10 values for non-temperature compensated biological reactions are generally in the range of 2.0-3.0 (Reyes et al., 2008). In contrast, the Q10 value reported for CPSF6 knock-down is 1.2 (Line 124, Fig. S2). Even if the difference in Q10 values between control and CPSF6 knock-down is statistically significant, this Q10 value is within the range of temperature compensation (0.85-1.2) as the authors mentioned (Lines 55-56 and 297-298). The subtle difference in Q10 could well be a statistical artefact without biological significance and it is far-fetched to conclude that CPSF6 knocked-down cells are resistant to temperature compensation, let alone "a tremendously impaired temperature compensation (line 322)". 

* Similarly, the Q10 value for EIF2S1 is also 1.2 and within the range of temperature compensation (Fig. S10), although statistical analysis was not performed for this dataset.

* It is unclear why the authors first screened siRNAs that would have different period or amplitude at 37C (Fig. S1) to understand the molecular mechanisms of temperature compensation. It would make more sense to test it at a different temperature. Please provide rationales for this.

* Even if CPSF6 was the regulator for temperature compensation, it is unclear why the authors hypothesized that the mRNA stability of core clock genes would be the primary mechanisms to regulate temperature compensation. There are many biological consequences of alternative polyadenylation, not just changes in mRNA stability. Additionally, CPSF6 has a broad role in 3'-end processing, not just alternative polyadenylation. 

* The authors should also briefly discuss what other mechanisms have been proposed to explain temperature compensation in mammalian cells (e.g., Beale et al., 2019, Shinohara et al., 2017, Isojima et al., 2009 etc)

* As the authors noted (Lines 182, 194-196, 210-211), there does not appear to be any novel findings from testing the effect of CPSF6 knock-down with APA, transcriptomic, and proteomic analyses performed at 37C. 

Minor points:

* Table S1 is incomprehensible. What does each column represent?

* Some figures are not mentioned in the text (e.g., Fig. 3D, 4D-F). The authors may want to remove these figures if they are not needed. The authors should also consider rearranging the order of figures to align with the main text. This would increase the readability of the manuscript and help future readers to better understand the study.

* Half-lives of mRNAs should be quantified to support the statement "we observed reduced mRNA half-lives (Fig. 1C)" (Line 139).

* Lines 116-117: please provide citations to support the statement that "thermosensitive regulation of pre-mRNA processing has been proposed as a mechanism underlying temperature compensation in other organisms".

Reviewer #2: In the manuscript titled "Alternative polyadenylation factor CPSF6 regulates temperature compensation of the mammalian circadian clock", by Schmal et. al address the mechanisms of circadian temperature compensation. Here, authors showed that knock-down of CPSF6 both increases circadian period length and temperature decompensates the circadian clock. Multi-omic analysis upon knock-down of CPSF6 confirms the CPSF6 affects 3' UTR length, as well as transcript and protein abundance of a large number of genes. Finally, the authors have suggested that EIF2S1 is a key player in the CPSF6 knock-down cells that also play a key role in temperature decompensation. 

The major contribution of the manuscript is identification of the alternative splice factor CPSF6 as a regulator of these two processes, and I think this point is made strongly. Mechanistic insights are not deep, however. The change in circadian period may well be due to increases in expression of many core genes. The change in temperature compensation is not well sorted out here; a number of candidates are proposed but not robustly validated, and no mechanistic insight is obtained.

The overall quality of the data is high, and the writing is clear (although see comments). 

Major comments:

1. Though authors showed that knock-down of CPSF6 leads to circadian period length and temperature decompensation, the mechanism of action of CPSF6 in circadian temperature compensation remains largely elusive and not deeply addressed. In page 16 authors listed out 4 different ways to address the mechanism of action of CPSF6, but these questions were not addressed in the manuscript. 

2. Authors identified EIF2S1 as a key player in temperature compensation from the CPSF6 knock-down cells using multi-omic approaches, but how these two molecules are connected in regulating circadian period and temperature compensation remains unknown and unaddressed. In fig. 5E second column, though there are differences in 3' UTR length and transcript of EIF2S1 in CPSF6 knock-down cells, the protein abundance remains almost the same in wild type and knock-down conditions. So, I think it is not clear whether EIF2S1 fulfill all the three criteria to be selected as a key player. 

3. It has been mentioned in the manuscript that either balance hypothesis or critical reaction hypothesis are the popular models for temperature compensation based on the previous publications. But in the current study, results support neither of these two models nor the authors propose a third alternative mechanism to explain mechanisms of temperature compensation, especially how CPSF6 alters the core clock machinery in lower and higher temperature conditions to do temperature compensation. 

Methodologic issues:

I believe that RNAseq can be quantitative and reproducible. I also believe (based on many experiences) that Mass spectrometry-based protein quantitation is MUCH less reliable than RNAseq. So, while I don't think it's important to validate RNAseq with qPCR, I do think it is critical to verify MS-identified changes in protein abundance using other methods, such as immunoblots.

A single RNAi against EIF2S1 is not adequate to draw conclusions. RNAi is rife with off-target effects. Studies that reply heavily on RNA interference (RNAi) to probe biological processes such as circadian rhythms must include appropriate controls such as those described in Nat Rev Drug Discov 9, 57-67 (2010). doi 10.1038/nrd3010. See specifically the paragraph 'Recognizing and confirming on-target effects.' Such controls must include use of multiple si/shRNAs for the same target, and/or (the best experiment) rescue of the phenotype by expression of target sequences refractory to si/shRNA.

Minor comments:

For the initial RNAi screen, how many siRNA per gene were used?

Table S1 (and others): there are many columns with cryptic headings. What do they mean? These should be defined. What is column 1? with no heading? What is "174"? 

Line 109: "CPSF6 is part of a multi-protein complex that is important for 3'-end processing of transcripts defining the polyadenylation site and thus the 3' UTR length." I wanted to learn more about CPSF6 and its known cellular functions. As a reader, I would have appreciated some references to papers about CPSF6.

NUDT21 is stated to be a heterodimeric binding partner of CPSF6. This deserves a reference as well. I found several sources stating that CPSF6 is in a tetramer, not a heterodimer. Which is correct?

116: "thermosensitive regulation of pre-mRNA processing has been proposed as a mechanism underlying temperature compensation in other organisms" should be referenced.

Figure S3 and others: the figure legends would be easier to read if it didn't refer to a different figure legend. I suggest putting much more information into the figure legend.

Triptolide is mentioned only once, in the figure legend. I don't know this compound, but based on the context I guess it is inhibiting RNA transcription. In the literature, it seems very toxic to cells. What is it supposed to be doing? Where did they buy it? What is the cellular IC50 of triptolide and what concentration was used in these assays? 

Are the differences in fig 1C and S3B statistically significant? In S3B, it appears n=1 in many cases. Can the authors justify why they are showing n=1 data (e.g. for NR1D1, PER1, PER2?).

Is the increased stability of mRNA of multiple core clock genes after CPSF6 knockdown (Fig S3) responsible for the longer period? Certainly tau, which decreases stability of PERs, decreases period, and increased stability of PER2, increased period length.

What are the units shown in Table S8? (note I didn't go through all the tables, but I think all were pretty sparse in explanations).

Reviewer #3: In chronobiology the regulation of circadian period length and its temperature compensation remain major unsolved issues. The authors performed a screen in U2OS cells for genes involved in post-transcriptional regulation and RNA processing which affect period length and temperature compensations. Out of 1024 genes, they identified that CPSF6 knockdown lengthens the period, and then showed that the compensation is also reduced (Q10 increased to 1.19 in U2OS). As CPSF6 is involved in 3'-end processing of transcripts and polyadenylation (PA) site selection the authors then combined 3-end transcriptomics and proteomics to identify the mechanism leading to the period and compensation phenotype wild-type and CPSF6 knockdown cells. While the knockdown induced many differential change in PA sites, mRNA and protein expression, the changes found upon different ambiant temperatures were largely independent of CPSF6. An integrative analysis of all the datasets to select for candidate genes that fulfill the characteristics of a regulator of temperature compensation led to the identification and testing (siRNA) of EIF2S1. Overall this is a well executed study with a large amount of new datasets, which overall sheds light on the complex CPSF6 biology, and how CPSF6 is implicated in period temperature compensation, possibly through EIF2S1 (aka eIF2α). While temperature compensation for the circadian period is usually thought to stem from the properties of core clock proteins, it is a new concept that global regulators such as CPS6 might also play a major role. Therefore the finding are novel and will be of great interest for PLoS Biology readers. 

Comments

1. Fig S1. It is not quite clear where the dose-dependent lengthening is shown. This may be an issue of the caption or axes labels (do the labels refer to the three constructs or doses?). Also the authors mention that period is similarly lengthened in 3T3 cells, which is important, but those results seems not as clear with respect to the doses? Do the authors have data from 3T3 cells that recapitulate the temperature compensation in U2OS in Fig. 1B? Since this is the key experiments that links with temperature compensation, this would be important to show.

2. Are the shorter half-lives and lower expression levels of the reported clock genes (Clock, Nr1d2) consistent with longer period? I.e. is the partial knockdown of those genes giving the same phenotype (see for example Baggs PLoS Biology 2009)? Or perhaps the authors have their own data on this?

3. Half-lives measurements. To which isoforms do the measurements refer to? Is there a way to distinguish and would this matter for the potential mechanism of period compensation?

4. Overall the authors have not used the length differences of the distal/proximal 3'UTR isoforms in their analysis. Based on the function of the polyadenylation complex, one would expect to see dependencies with length in several variables (%usage, etc). 

5. Could the authors elaborate further on the relation between mRNA and protein levels? Is this global correlation, or lack of correlation in certain cases, excepted and comparable to other datasets? What are the reasons? Are there certain classes of proteins that show more, or less correlation, etc. 

Minor

In general, is it typical that period mutants also show temperature compensation defects? The authors might discuss some striking examples.

Fig 1C: First and second rows. Are the changes in relative mRNA expression consistent with the total mRNA expression bars (lower row, middle panel)?

"our data suggest that processes other than mRNA stability regulate changes of PER3 expression." Could this be true also for the other clock genes, when consider the quantitative changes of half-lives sand expression levels?

It is unclear if normalizing the NCOM to the length of the 3' UTR is always a good thing as it might hide some length dependent effects which might be expected.

Fig. 2B. Scatter plot of one vs other might be useful as well.

In https://pubmed.ncbi.nlm.nih.gov/33648552/

"CPSF6 links alternative polyadenylation to metabolism adaption in hepatocellular carcinoma progression.", there is also a RNA-seq mapping of PASs upon knockdown. The authors should refer to it.

---

## [Decision Letter · Decision Letter 2]

17 Apr 2023

Dear Dr Schmal,

Thank you for your patience while we considered your revised manuscript "Alternative polyadenylation factor CPSF6 regulates temperature compensation of the mammalian circadian clock" for publication as a Research Article at PLOS Biology. This revised version of your manuscript has been evaluated by the PLOS Biology editors, the Academic Editor, and the original reviewers.

The reviews are appended below and as you will see the reviewers appreciate the substantial effort that has gone into this revision and Reviewers 2 and 3 have suggested we accept the study. After careful discussion with the Academic Editor, we think that the current revision has addressed the reviewer comments to our satisfaction and were are therefore likely to accept the study. However, we note that Reviewer 1 has a number of lingering concerns and additional points that may require further consideration. We therefore wanted to give you the chance to read and respond to Reviewer 1’s remaining reservations, and we encourage you to adopt any changes to the manuscript related to these points that you think would improve the study.

**IMPORTANT: Additionally, before we can accept your manuscript for publication, we have a number of editorial requests which we need you to address:

1) BLURB: Please provide a blurb which (if accepted) will be included in our weekly and monthly Electronic Table of Contents, sent out to readers of PLOS Biology, and may be used to promote your article in social media. The blurb should be about 30-40 words long and is subject to editorial changes. It should, without exaggeration, entice people to read your manuscript. It should not be redundant with the title and should not contain acronyms or abbreviations.

2) DATA REQUEST: Thank you for uploading your sequencing and proteomic data to data repositories. I tried to access the proteomic dataset on the ProteomeXchange Consortium via the PRIDE (https://www.ebi.ac.uk/pride/archive ; username: reviewer_pxd029343@eb i.ac.uk and password: bOWYvyOk), however I did not see the dataset (PXD029343). Can you please double check that the information to access this dataset in your data availability statement is correct?

3) DATA REQUEST: You may be aware of the PLOS Data Policy, which requires that all data be made available without restriction: http://journals.plos.org/plosbiology/s/data-availability. For more information, please also see this editorial: http://dx.doi.org/10.1371/journal.pbio.1001797

While the deposition of the sequencing and proteomic datasets meets this requirements for most of your figure panels - for any figures not related to those datasets, we need you to provide the underlying data. (maybe Fig 1B_C; Fig S1, S2, S11; Supp text figure 4).

a - Supplementary files (e.g., excel). Please ensure that all data files are uploaded as 'Supporting Information' and are invariably referred to (in the manuscript, figure legends, and the Description field when uploading your files) using the following format verbatim: S1 Data, S2 Data, etc. Multiple panels of a single or even several figures can be included as multiple sheets in one excel file that is saved using exactly the following convention: S1_Data.xlsx (using an underscore).

b - Deposition in a publicly available repository. Please also provide the accession code or a reviewer link so that we may view your data before publication.

Regardless of the method selected, please ensure that you provide the individual numerical values that underlie the summary data displayed in the following figure panels as they are essential for readers to assess your analysis and to reproduce it.

>>>Please also ensure that figure legends in your manuscript include information on where the underlying data can be found, and ensure your supplemental data file/s has a legend. For example, to each figure legend (including supplemental) you can add the sentence "the data underlying this figure is available from the NCBI Gene Expression Omnibus (GEO; https://www.ncbi.nlm.nih.gov/geo/) under accession number GSE185896, the ProteomeXchange Consortium via the PRIDE partner repository with the dataset identifier PXD029343, or is contained in S1_Data."

4) GELS AND BLOT REPORTING: We require the original, uncropped and minimally adjusted images supporting all blot and gel results reported in an article's figures or Supporting Information files. We will require these files before a manuscript can be accepted so please prepare and upload them now. Please carefully read our guidelines for how to prepare and upload this data: https://journals.plos.org/plosbiology/s/figures#loc-blot-and-gel-reporting-requirements

>>Please provide the uncropped and unadjusted gel related to Supp Text Fig 2 (page gel) - annotated as described in the guidelines above.

We expect to receive your revised manuscript within two weeks, although do let us know if you need more time to complete the revision. 

*Published Peer Review History*

*Press*

Sincerely,

Luke

Lucas Smith, Ph.D.

Associate Editor,

lsmith@plos.org,

PLOS Biology

Reviewer Comments: 

Reviewer #1: The revised manuscript by Schmal et al., is improved and addressed many of my original comments. Because of this, the authors are strongly encouraged to include additional data and text in the response letter to the main manuscript, as some data are integral to support the validity of the study. Below, I describe a few remaining reservations. I also added further thoughts based on new information provided in the revised version. The authors addressed all the other comments.

Responses to the major points

1. The authors provided additional data, in which the efficiency of knock-down was evaluated at three different temperatures. The data are an important piece of information to support the authors' conclusion and hence should be added to the manuscript. At the same time, the data, as currently presented, lack critical information: a) what is 100 in Y-axis? Relative to what?, b) how many independent tests? C) what was the amount (or concentration) of shRNAs in reference to Fig. 1A-B, or Fig. S1A-B?

2. As I believe there is no clear threshold for both absolute Q10 values and relative changes (deltaQ10) to define whether the process is temperature compensated or not, I am fine with leaving this point up to the post-publication peer-review by a broader scientific community. To execute this well, however, the authors should provide all the information available for the readers so that they can make their own judgment calls. 

There are many papers that focus on temperature compensation. Some of these papers calculated Q10 values, while most did not (which actually surprised me!). There appears to be a range of absolute and relative Q10 values, including those similar to the values reported in this manuscript, used to claim whether the process is temperature compensated or not. The authors should succinctly but comprehensively summarize the previous findings and compare those with their finding (also see comments below). The authors picked one paper (Huang et al., Science 1995) and calculated the Q10 value, however, it is not justifiable to cite one paper to support the significance of the authors' findings. 

Other examples (that I was able to find):

https://doi.org/10.1128/mBio.01425-21

https://www.jstor.org/stable/4267082

https://doi.org/10.3389/fphys.2022.888262

https://doi.org/10.1016/j.isci.2020.101388

https://doi.org/10.1177/0748730415613888

https://doi.org/10.1371/journal.pcbi.1005501

https://doi/10.1126/sciadv.abe8132 (cited but not in this context)

https://doi.org/10.1016/j.molcel.2015.08.022 (cited but not in this context)

https://doi.org/10.1073/pnas.0908733106 (cited but not in this context)

To clarify, I am not requesting the authors calculate Q10 values of all of these studies. Rather, I am requesting the authors provide all the information so that readers can be fully informed to draw their own conclusions. 

This is the authors' statement: "circadian free-running periods have been found to be remarkably resilient to changes in ambient temperature, typically exhibiting Q10 temperature coefficients between 0.85 to 1.2 (lines 22-24)". Meanwhile, their major finding in this manuscript is that Q10 values are different between control and Cpsf6 KD (Ctrl: 1.1 vs Cpsf6 KD: 1.2), but both values are within the range of temperature compensation defined by the authors (based on citations). I am simply confused - if there are changes in Q10 values between different conditions, even if the absolute Q10 values are within the range of "temperature compensation (e.g., 0.85-1.2)", should that be considered as temperature de-compensation or under-compensation? What changes are small enough to say that it is NOT temperature compensated if statistics support the difference? Please summarize current consensus in the field and discuss this further.

4. (This is also related to the response to the last minor point): The authors mentioned in the response letter that they designed the experiments based on "prior knowledge from previous studies on temperature compensation that found post-transcriptional mechanisms to play a major role in temperature compensation". Please provide references for these studies. I checked the references 35, 36, 37, 38, 39, and 41, and these are either describing thermo-sensitive alternative splicing (not temperature compensation) or roles of casein kinase (post-translational, not post-transcriptional, regulator) in regulating temperature compensation. I agree that thermo-sensitive alternative splicing may be the mechanism to detect the difference in temperature, but I was not able to find any papers demonstrating that alternative splicing (or other pre-mRNA processing processes) is a mechanism underlying temperature compensation. Or post-transcriptional mechanisms playing a 'major' role in temperature compensation. As the authors are probably aware, changes in period are not the same as temperature compensation (refs 43, 44).

Other new points:

Do the authors really think CPSF6 is important for temperature compensation? There are some statements in the manuscript that suggest otherwise (e.g., lines 283-284, 393-398). In addition, temperature compensation is one of the main characteristics in defining circadian rhythms observed in almost all living organisms on Earth, while alternative polyadenylation is primarily a mechanism for eukaryotic gene expression. The authors also mentioned alternative polyadenylation is highly tissue-specific (lines 437-482). Do the authors think that CPSF6-mediated temperature compensation (if true) is tissue-dependent? Do multicellular organisms have some tissues temperature compensated and other tissues not? How do these organisms exhibit temperature compensation at an organismal level if the regulation is tissue-specific? The authors should reconcile these statements and discuss this further. 

Along the same line, the authors should also discuss how their new findings align with other mechanisms that have been already proposed (i.e., CK-mediated regulation). Do the authors think that either or both is correct or incorrect? Why or why not?

Very minor:

Lines 378-379: There is no data showing that the KD of CSPF6 leads to "a significantly altered temperature compensation" in NIH3T3 cells. Changes in period at one temperature is not the same as the system being temperature compensated.

Reviewer #2: The manuscript is improved by the many changes. The core observation, that CPSF6 regulates the core circadian clock, remains robust. This revision, which clearly required substantial additional time and effort, unfortunately was not able to successfully address additional mechanistic questions raised by this interesting observation. 

Reviewer #3, Felix Naef (note, reviewer 3 has signed this review): The authors have provided highly thorough and convincing answers to all of my points. Thank you for adding new panels to Figure S5. I highly recommend rapid publication in PLoS Biology.

Felix Naef

---

## [Editor Report · Decision Letter 3]

15 May 2023

Dear Dr Schmal,

Thank you for the submission of your revised Research Article "Alternative polyadenylation factor CPSF6 regulates temperature compensation of the mammalian circadian clock" for publication in PLOS Biology. On behalf of my colleagues and the Academic Editor, Samer Hattar, I am pleased to say that we are satisfied by the changes made in the most recent revision and can in principle accept your manuscript for publication, provided you address any remaining formatting and reporting issues. These will be detailed in an email you should receive within 2-3 business days from our colleagues in the journal operations team; no action is required from you until then. Please note that we will not be able to formally accept your manuscript and schedule it for publication until you have completed any requested changes.

PRESS

Sincerely, 

Lucas Smith, Ph.D.

Associate Editor

PLOS Biology

lsmith@plos.org